# Gut microbiome signatures of vegan, vegetarian and omnivore diets and associated health outcomes across 21,561 individuals

Gloria Fackelmann [1], Paolo Manghi [1], Niccolò Carlino [1], Vitor Heidrich [1], Gianmarco Piccinno[1], Liviana Ricci [1], Elisa Piperni [1], Alberto Arrè[2], Elco Bakker [2], Alice C. Creedon [2], Lucy Francis[2], Joan Capdevila Pujol[2], Richard Davies [2], Jonathan Wolf[2], Kate M. Bermingham[2], Sarah E. Berry[3,6], Tim D. Spector [2,4,6], Francesco Asnicar [1,6] & Nicola Segata [1,4,5,6] ✉

As plant-based diets gain traction, interest in their impacts on the gut microbiome is growing. However, little is known about diet-pattern-specific metagenomic profiles across populations. Here we considered 21,561 individuals spanning 5 independent, multinational, human cohorts to map how differences in diet pattern (omnivore, vegetarian and vegan) are reflected in gut microbiomes. Microbial profiles distinguished these common diet patterns well (mean AUC = 0.85). Red meat was a strong driver of omnivore microbiomes, with corresponding signature microbes (for example, *Ruminococcus torques*, *Bilophila wadsworthia* and *Alistipes putredinis*) negatively correlated with host cardiometabolic health. Conversely, vegan signature microbes were correlated with favourable cardiometabolic markers and were enriched in omnivores consuming more plant-based foods. Diet-specific gut microbes partially overlapped with food microbiomes, especially with dairy microbes, for example, *Streptococcus thermophilus*, and typical soil microbes in vegans. The signatures of common western diet patterns can support future nutritional interventions and epidemiology.

Diet is inextricably linked to human health. Globally, poor diets low in unprocessed, plant-based foods cause more deaths than any other risk factor, with cardiovascular disease, cancers and type 2 diabetes as the leading causes of diet-related deaths[1]. Unhealthy diets also carry a wide range of negative environmental impacts[2]. Animal-based foods contribute comparably more than plant-based foods to global environmental change through their impact on climate, land and freshwater use, and biodiversity[2,3]. Consequently, there is increased interest in diets with higher fractions of plant-based foods that decrease both risk of disease and negative environmental impacts[2,4].

The gut microbiome plays an integral role in human health that can be modified by diet[5]. For example, fermentation of otherwise indigestible plant polysaccharides by gut microbes contributes to a healthy, non-inflamed gut barrier and maintenance of gut homoeostasis through the production of short-chain fatty acids (SCFAs) and immune system crosstalk[6]. Moreover, plants contain polyphenols, the

[1]Department of Cellular, Computational and Integrative Biology, University of Trento, Trento, Italy. [2]ZOE Ltd., London, UK. [3]Department of Nutritional Sciences, King's College London, London, UK. [4]Department of Twins Research and Genetic Epidemiology, King's College London, London, UK. [5]European Institute of Oncology, Scientific Institute for Research, Hospitalization and Healthcare, Milan, Italy. [6]These authors jointly supervised this work: Sarah E. Berry, Tim D. Spector, Francesco Asnicar, Nicola Segata. ✉e-mail: nicola.segata@unitn.it

**Fig. 1 | A large, integrated, metagenomic dataset with detailed dietary information. a,** Sample size for each diet pattern across the five cohorts (logarithmic scale). **b,** Observed richness of each diet pattern's gut microbiome within each of the five cohorts ($n_{P1}$ = 1,062 individuals, $n_{P3\,UK22A}$ = 12,353, $n_{P3\,US22A}$ = 7,931, $n$ Tarallo et al. (2022)[12] = 118, $n$ De Filippis et al. (2019)[13] = 97). Boxplots show the median, 25th and 75th percentiles, and whiskers extend to 1.5× the interquartile range. Asterisks denote significance level of Dunn's tests with BH correction (Methods and Supplementary Table 6); *$P$ < 0.05, **$P$ ≤ 0.01, ***$P$ ≤ 0.001. **c,** Distribution of hPDI for each diet pattern within each of the five cohorts (P1 with 841 omnivores, 49 vegetarians and 10 vegans; P3 UK22A with 11,289 omnivores, 610 vegetarians and 192 vegans; P3 US22A with 6,720

omnivores, 309 vegetarians and 346 vegans). Boxplot integrated into violin plots have the same parameters as in **b**. Asterisks denote the same significance as in **b**, but for Tukey contrasts for multiple comparisons following an ANOVA model (Methods and Supplementary Table 3). **d–h,** Beta diversity of gut microbial composition accounting for phylogenetic diversity using unweighted UniFrac distances. Each dot in the principal coordinates analysis (PCoA) plots represents an individual. Ellipses indicate 95% CIs. Statistical differences between diet patterns were assessed via PERMANOVA, correcting for sex, age and BMI with 999 permutations. There is one PCoA plot per cohort: P1 (**d**), P3 UK22A (**e**), P3 US22A (**f**), De Filippis et al. (2019)[13] (**g**), Tarallo et al. (2022)[12] (**h**).

products of plant secondary metabolism, that are known to promote beneficial bacteria that prevent inflammation, enhance the gut barrier and hinder potential pathogens[7].

By contrast, a diet rich in animal foods leads to increased protein fermentation, which may result in a leaky mucosa, local and systemic inflammation and reduced production of SCFAs[8]. For example, the breakdown of certain animal proteins is linked to the synthesis of gut microbial trimethylamine (TMA), which is oxidized in the liver to trimethylamine N-oxide (TMAO)[6]. TMAO has been implicated in various (cardio)vascular diseases and is a potential contributing factor in colorectal cancer[9]. However, both dietary information and gut microbiome composition are extremely variable and noisily surveyed, and the current state-of-the-art in diet–gut microbiome links lacks a large-scale, cross-country and cross-cohort approach able to disentangle more nuanced associations between particular dietary aspects and individual gut microbes at the species level. Currently, health associations use the same basic datasets for vegans and omnivores, despite large potential baseline differences and biases.

## Results

### Multicohort gut metagenomics with detailed dietary data
The aim of this study was to elucidate how prolonged dietary preferences affect the structure and function of the human gut microbiome at both the global and single-species level. To do so, we capitalized on three cohorts from the ZOE PREDICT programme from the United Kingdom (P1 $n$ = 1,062 individuals[10,11], P3 UK22A $n$ = 12,353) and from the United States (P3 US22A $n$ = 7,931; Methods). We further included two additional, publicly available cohorts comprising Italian participants

(Tarallo et al. (2022)[12] $n$ = 118 individuals and De Filippis et al. (2019)[13] $n$ = 97 individuals; Fig. 1a). Each participant of the five cohorts reported their nutritional habits as being either 'omnivore' (including meat, dairy and vegetables), 'vegetarian' (excluding meat) or 'vegan' (excluding both meat, dairy and other animal products) and donated stool samples that underwent shotgun metagenomic sequencing. In total, 656 vegans, 1,088 vegetarians and 19,817 omnivores were considered (Fig. 1a). In addition to participants' overall dietary habits, the ZOE PREDICT cohorts included data on habitual consumption of over 150 single foods per individual, obtained from validated quantitative food frequency questionnaires (FFQs; Methods). Dietary patterns were partially confirmed by DNA-based detection of food in the stool microbiome[14] which, however, would require greater sequencing depth to be used for this goal (Methods).

To quantify the consumption of plant-based foods, we considered the plant-based diet index (hPDI), which gives higher scores to healthy plant foods and reverse scores to less healthy plant and animal foods[15] (Methods). Within each of the three PREDICT cohorts, hPDI significantly differed between diet patterns as expected (analysis of variance (ANOVA), $P$ < 0.001 across all PREDICT cohorts; Fig. 1c and Supplementary Table 1), with significantly higher hPDI in vegans compared with vegetarians and similarly for vegetarians compared with omnivores (Tukey $P$ < 0.01; Supplementary Tables 2 and 3).

### Gut microbial diversity and composition across diet patterns
Gut microbial richness differed significantly according to diet patterns in the PREDICT cohorts (Kruskal–Wallis, $P$ < 0.05; Supplementary Table 4), with a lower observed richness in vegans (median between

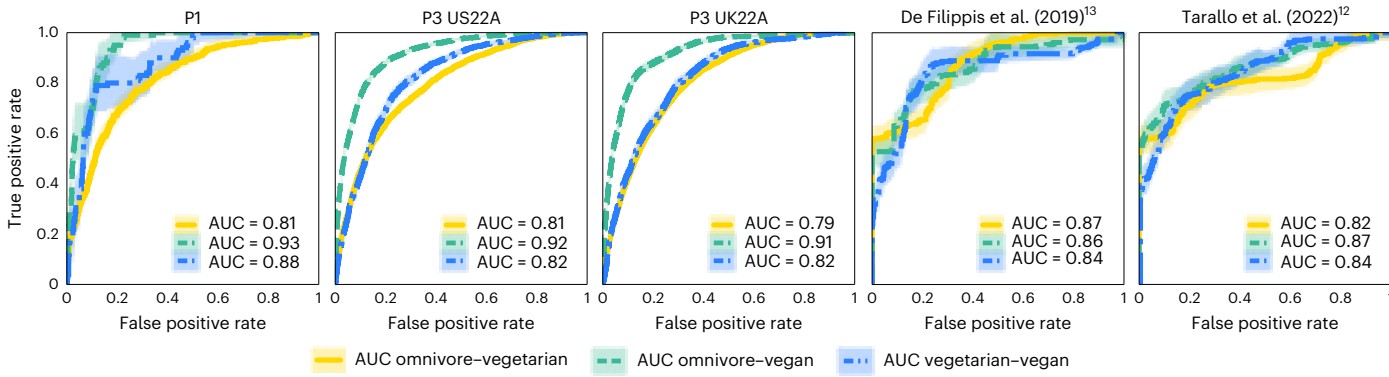

**Fig. 2 | Highly accurate classification of individual diet patterns based on gut microbial features.** Average ROC curves and AUCs showing the discrimination between all pairs of the three diet patterns (omnivores vs vegetarians; omnivores vs vegans; and vegetarians vs vegans) per cohort using random forest classifiers in a hybrid cross-LODO (Methods) approach. Shaded areas correspond to 95% CIs.

209 and 266 species-level genome bins (SGBs)) and vegetarians (median 201–269) compared with omnivores (median 217–299; Fig. 1b and Supplementary Table 5), but no significant differences between vegans and vegetarians (Dunn's test, $P > 0.05$; Supplementary Tables 6 and 7). This highlights that alpha diversity might correlate with diet patterns that are potentially more diverse.

Overall gut microbial composition also differed significantly according to diet pattern (permutational multivariate analysis of variance (PERMANOVA) on unweighted UniFrac distances, $R^2 = 0.002$– 0.028; $P < 0.05$ for all five cohorts; Fig. 1d–h and Supplementary Table 8 with additional distance metrics; Methods), with the variation in beta diversity explained by diet pattern aligning with previous studies[16]. In addition, diet patterns were highly distinguishable based on quantitative gut microbial profiles when using machine learning classifiers[17]. By evaluating the performance of the model trained in a variant of cross validation in which training folds are merged with external cohorts (cross-validation leave-one-dataset-out, that is, cross-LODO; Methods)[18], we obtained a mean area under the receiver operating characteristic (ROC) curve (AUC) across all diet patterns and across all five cohorts of 0.85. The highest predictability was obtained when separating vegans from omnivores (mean cross-LODO AUC = 0.90), followed by separating vegetarians from vegans (0.84), and finally vegetarians from omnivores (0.82; Fig. 2 and Supplementary Table 9). Similar results were achieved when using the LODO approach that does not consider any training folds from the target cohort (Supplementary Table 9). Because we did not log when diet patterns may have been switched, we hypothesize that the non-perfect classification might be due to individuals who switched diet patterns recently, and some associations may actually be stronger than what we observed. Altogether, these results warranted further investigation into the specific microbiome components responsible for these differences.

**Gut microbe signatures of vegans, vegetarians and omnivores**

To explore which microbes are associated with the different gut microbial compositions between vegans, vegetarians and omnivores, we performed a meta-analysis across the five cohorts on the differential relative abundance of each SGB within each individual and their respective diet pattern (Methods). In total, 488 SGBs were significantly differentially abundant in omnivores compared with 112 SGBs in vegetarian microbiomes; 626 SGBs were significantly differentially abundant in omnivores compared with 98 SGBs in vegans; and 30 SGBs were significantly differentially abundant in vegetarins compared to 11 SGBs in vegans (Supplementary Tables 10–12). When focusing on the top 30 microbial markers, the majority of these strongest associations were linked to the least restrictive diet pattern (Figs. 3b,h and 4b).

Knowledge of the predicted functions of the SGBs linked to the various diet patterns revealed potential dietary-specific niches. Several SGBs increased in omnivore microbiomes are linked to meat consumption by aiding in its digestion through for example, protein fermentation (*Alistipes putredinis*), utilizing amino acids and via bile-acid resistance (*Bilophila wadsworthia*[19]), or are mucolytic indicators of inflammation that have been linked to inflammatory bowel diseases (*Ruminococcus torques*[20,21]; Fig. 3b,h). In contrast, several SGBs over-represented in vegan microbiomes are known butyrate producers (Lachnospiraceae[22], *Butyricicoccus* sp.[23,24] and *Roseburia hominis*[22,25]) and are highly specialized in fibre degradation (Lachnospiraceae[26]; Figs. 3h and 4b). In addition, *Streptococcus thermophilus*, a common dairy starter and component[27], had the highest effect size in vegetarian versus vegan gut microbiomes with a standardized mean difference (SMD) of −0.67 and second highest effect size in omnivore versus vegan gut microbiomes (SMD = −0.62). Thus, when a major differentiating characteristic between diet patterns lies in dairy consumption, the SGB with the greatest ability to differentiate between those diets is abundantly found in cheese and yogurt products. This was supported by other dairy-linked SGBs associated more with omnivore and vegetarian than vegan diets such as *Lactobacillus acidophilus*, *Lactobacillus delbrueckii*, *Lactococcus lactis*, *Lacticaseibacillus paracasei* and *Lacticaseibacillus rhamnosus*[27,28]. On the basis of these findings, we next explored the links between these diet pattern-specific microbes and the major food groups that distinguish the diet patterns.

**Gut microbial diet signatures are linked to major food groups**

We further investigated the role of major food groups, such as red and white meat, dairy, fruits and vegetables, in differentiating the gut microbial profiles across diet patterns (Methods and Supplementary Table 13). The amount of meat (either red or white) ingested by omnivores was positively correlated with the vast majority of SGBs linked to an omnivorous diet versus a vegetarian (23 out of 25 SGBs; Fig. 3c) or vegan one (16 of out 19 SGBs; Fig. 3i). In addition, compared with omnivore gut microbiomes, meat negatively correlated with all 5 SGBs strongly associated with vegetarian gut microbiomes and with 10 out of the 11 SGBs strongly associated with vegan gut microbiomes. The SGBs strongly associated with omnivore gut microbiomes correlated more strongly with red than with white meat consumption. Red and white meat correlated with the same SGBs in all but one case: '*Candidatus* Avimicrobium caecorum', found in human gut microbiomes and assembled from chicken caecum[29], which positively correlated with white meat consumption in omnivore gut microbiomes versus vegetarian and vegan ones (Fig. 3c,i).

In contrast, fruits and vegetables were positively correlated with 3 of out 5 SGBs overrepresented in vegetarian (Fig. 3c) and in 10 out

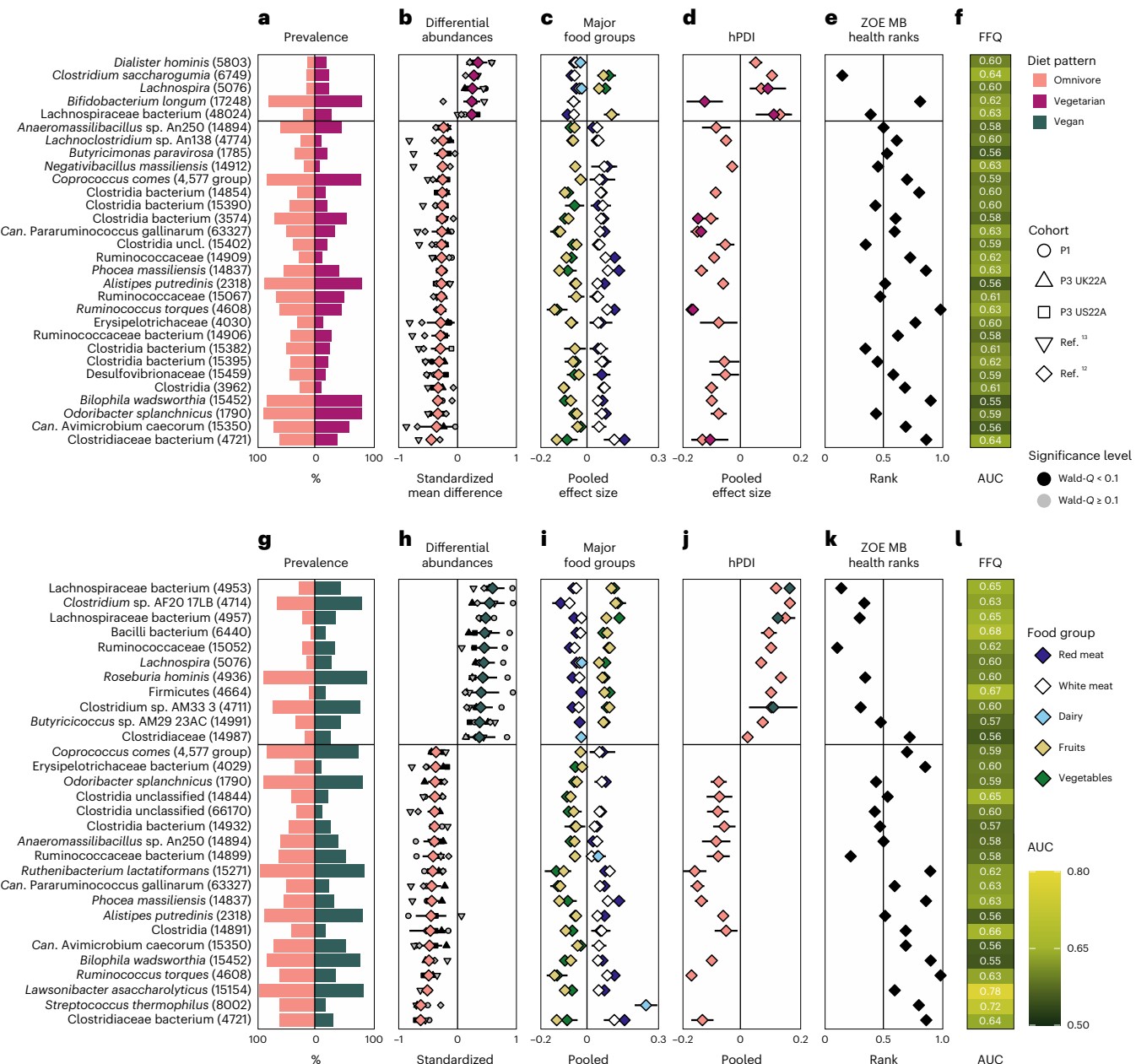

**Fig. 3 | Gut microbial signatures of an omnivore vs vegetarian and vegan diets.**
Top panels: omnivore vs vegetarian diet. Bottom panels: omnivore vs vegan
diet. **a**, Prevalence of the top 30 signature SGBs (with their respective SGB IDs in
parentheses) in omnivore (left) and vegetarian (right) gut microbiomes.
**b**, Meta-analysed correlations between SGB relative abundance and diet pattern
(omnivore $n = 19,817$ in pink vs vegetarian $n = 1,088$ in purple). The top 30 SGBs
with the largest absolute SMD are reported, with upper and lower confidence
intervals. Smaller shapes are per-cohort correlations (black indicates Wald $q$-
value < 0.1, grey indicates Wald $q$-value ≥ 0.1). The black horizontal bar indicates
the separation between the correlations with omnivores vs vegetarians for ease
of visualization only. **c**, Meta-analysed pooled effect sizes with upper and lower
confidence intervals from correlations between SGB relative abundance and
consumption of five major food groups (meat: $n_{P1} = 841$ individuals, $n_{P2} = 843$,
$n_{P3\,UK22A} = 11,533$, $n_{P3\,US22A} = 7,228$; dairy: $n_{P1} = 890$, $n_{P2} = 843$, $n_{P3\,UK22A} = 12,156$,
$n_{P3\,US22A} = 7,558$; fruits/vegetables: $n_{P1} = 900$, $n_{P2} = 843$, $n_{P3\,UK22A} = 12,353$,

$n_{P3\,US22A} = 7,931$). **d**, Meta-analysed pooled effect sizes with upper and lower
confidence intervals from correlations between SGB relative abundance and
hPDI within omnivores ($n_{P1} = 841$, $n_{P2} = 843$, $n_{P3\,UK22A} = 11,289$, $n_{P3\,US22A} = 6,720$)
and vegetarians ($n_{P1} = 49$, $n_{P3\,UK22A} = 610$, $n_{P3\,US22A} = 309$). **e**, ZOE MB health ranks
of each signature SGB. Values closer to zero indicate positive CMH outcomes,
closer to one indicate negative CMH outcomes[30]. **f**, Machine learning predictions
(random forest cross-LODO AUC; Methods) of the presence of each of the
signature microbes between omnivores and vegetarians based on FFQs.
**g**, Prevalence of the top 30 signature SGBs (with their respective SGB IDs in
parentheses) in omnivore (left) and vegan (right) gut microbiomes. **h**, Same
as **b**, except between omnivores ($n = 19,817$ in pink) and vegans (n = 656 in green).
**i**, Same as **c**, except between omnivores and vegans. **j**, Same as **d**, except
between omnivores and vegans (vegans: $n_{P1} = 10$, $n_{P3\,UK22A} = 192$, $n_{P3\,US22A} = 346$).
**k**, Same as **e**, except between omnivores and vegans. **l**, Same as **f**, except between
omnivores and vegans.

of 11 SGBs overrepresented in vegan versus omnivore gut microbi-
omes (Fig. 3i). The majority of these correlations were more greatly
associated with vegetables than with fruits. There were no cases of
negative correlations between fruits and vegetables and the SGBs

most strongly associated with vegetarian or vegan gut microbiomes.
Conversely, any SGB strongly linked to an omnivore gut microbiome
that correlated with fruits or vegetables showed negative and not
positive correlations.

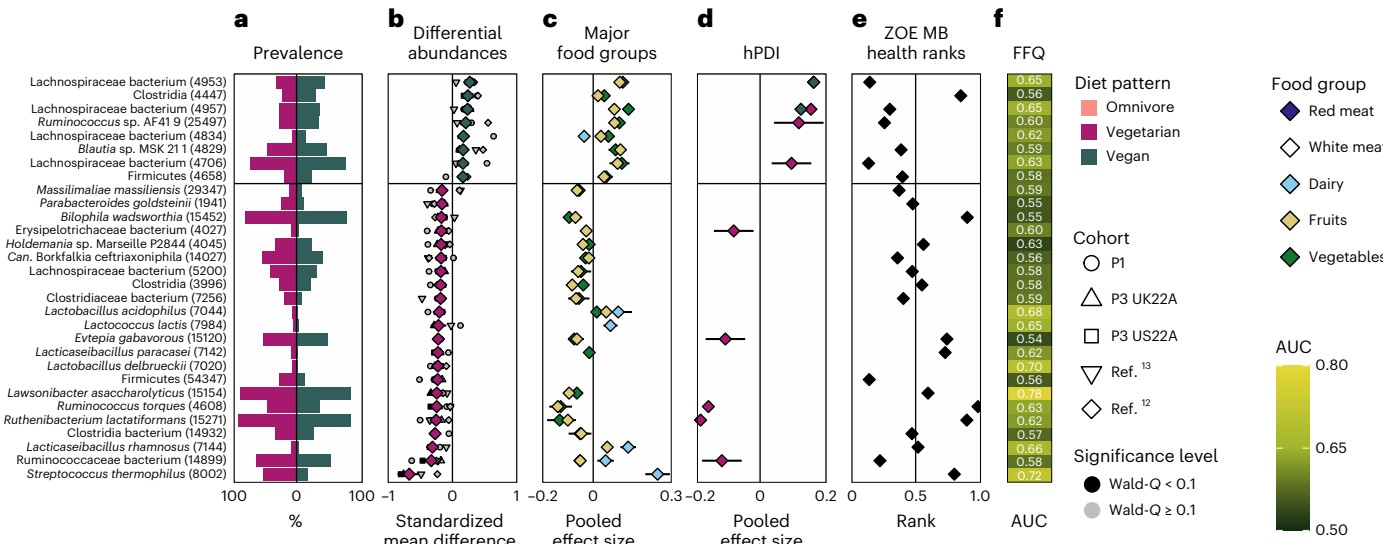

**Fig. 4 | Gut microbial signatures of a vegetarian vs vegan diet. a**, Prevalence of the top 30 signature SGBs (with their respective SGB IDs in parentheses) in vegetarian (left) and vegan (right) gut microbiomes. **b**, Meta-analysed correlations between SGB relative abundance and diet pattern ($n_{vegetarian} = 1,088$ in purple vs $n_{vegan} = 656$ in green). The top 30 SGBs with the largest absolute SMD are reported. Smaller shapes are per-cohort correlations (black indicates Wald $q$-value < 0.1, grey indicates Wald $q$-value ≥ 0.1). The black horizontal bar indicates the separation between the correlations with vegetarians vs vegans for ease of visualization only. **c**, Meta-analysed pooled effect sizes with upper and lower confidence intervals from correlations between SGB relative abundance and consumption of five major food groups (meat: $n_{P1} = 841$ individuals,

$n_{P2} = 843$, $n_{P3UK22A} = 11,533$, $n_{P3US22A} = 7,228$; dairy: $n_{P1} = 890$, $n_{P2} = 843$, $n_{P3UK22A} = 12,156$, $n_{P3US22A} = 7,558$; fruits/vegetables: $n_{P1} = 900$, $n_{P2} = 843$, $n_{P3UK22A} = 12,353$, $n_{P3US22A} = 7,931$). **d**, Meta-analysed pooled effect size with upper and lower confidence intervals from correlations between SGB relative abundance and hPDI within vegetarians ($n_{P1} = 49$, $n_{P3UK22A} = 610$, $n_{P3US22A} = 309$) and vegans ($n_{P1} = 10$, $n_{P3UK22A} = 192$, $n_{P3US22A} = 346$). **e**, ZOE MB health ranks of each signature SGB. Values closer to zero indicate positive CMH outcomes, closer to one indicate negative CMH outcomes. **f**, Machine learning predictions (random forest cross-LODO AUC; Methods) of the presence of each of the signature microbes between vegetarians and vegans based on FFQs.

When considering dairy, which differentiates vegans from vegetarians and contributes to the difference between a vegan and an omnivore diet, SGBs that differentiate vegetarian from vegan gut microbiomes showed positive correlations with dairy in vegetarians and negative ones in vegans (Fig. 4c). Similarly, SGBs that differentiate omnivore from vegan gut microbiomes showed positive associations with dairy in omnivores and negative ones in vegans (Fig. 3i). Thus, the gut microbial signatures of these three diet patterns are linked to the inclusion or exclusion of major food groups.

**Plant-based food diversity shapes the microbiome across diets**
While the three diet patterns differed significantly in their hPDI scores (Fig. 1c), we next aimed to understand whether their correlations with the SGB relative abundance were consistent across diet patterns using a meta-analytical approach (Methods). Regardless of which diet patterns were compared, there was concordance in the correlations between hPDI and the SGB signature of each diet pattern (Figs. 3d,j and 4d). This means that if hPDI was correlated (either positively or negatively) with a signature SGB in omnivore gut microbiomes, it would show similar correlations in vegetarians and vegans as well. Thus, overall dietary factors may transcend diet patterns, suggesting that omnivores could share beneficial gut microbial signatures with other diet patterns if they also incorporate similar diversity of plant-based food items in their diets. In practice, however, omnivores generally ingest significantly less healthy plant-based foods than vegetarians or vegans (Fig. 1c).

**Cardiometabolic health is linked to gut microbial diet patterns**
To investigate the gut microbial links between the three diet patterns and human health, we employed the ZOE Microbiome Ranking 2024 (Cardiometabolic Health), ZOE MB Health ranks for short[30], which assigns a numeric ranking to SGBs found to significantly correlate with cardiometabolic markers (Methods). We found that rankings of SGB signatures of omnivore microbiomes were statistically less favourable

(mean rank = 0.53 and 0.58) when compared with vegetarian (mean rank = 0.44, two-sample $t$-test, $P = 0.040$, $t(197) = 2.07$) and vegan ones (mean rank = 0.38, two-sample $t$-test, $P < 0.001$, $t(230) = 5.59$; note that values closer to zero indicate positive CMH outcomes, whereas values closer to one indicate negative CMH outcomes; Extended Data Fig. 1). When comparing rankings of SGB signatures of vegan versus vegetarian microbiomes, vegan-associated SGBs had more favourable rankings (mean rank = 0.33) than vegetarian-associated ones (mean rank = 0.54, two-sample $t$-test, $P = 0.028$, $t(30) = 2.30$; Extended Data Fig. 1). These patterns were reflected when considering the 30 SGBs most distinguishable between the diet patterns. The majority of the ranked SGB signatures of an omnivore gut microbiome were associated with worse cardiometabolic health (CMH) compared with both vegetarian and vegan gut microbiomes, with the opposite being true for vegetarian and vegan gut microbiomes (Fig. 3e,k). When comparing vegetarian with vegan gut microbiomes, the latter again showed a majority of signature SGBs to be associated with positive CMH, whereas the pattern for the former was more split, with just under half of the vegetarian signature SGBs linked with more favourable CMH (Fig. 4e). Thus, omnivore signature microbes are associated with less favourable CMH, whereas signature vegan microbes are associated with more favourable CMH.

**Entire diet profiles can predict specific gut species**
Moving from major food groups to the entire set of food items in the FFQs, we next tested the extent to which habitual-diet information is linked to the presence or absence of each SGB of relevance for the three diet patterns (Methods[18]). The most diet-linked SGBs were those that most differentiate between omnivore and vegan gut microbiomes, in particular *S. thermophilus*, predictable from whole FFQ items at AUC = 0.72, *R. torques* (0.63), several Lachnospiraceae SGBs (all 0.65) and *Lawsonibacter asaccharolyticus* (0.78; Figs. 3f,l and 4f, and Supplementary Tables 14–16), which is strongly tied to coffee consumption[10,31].

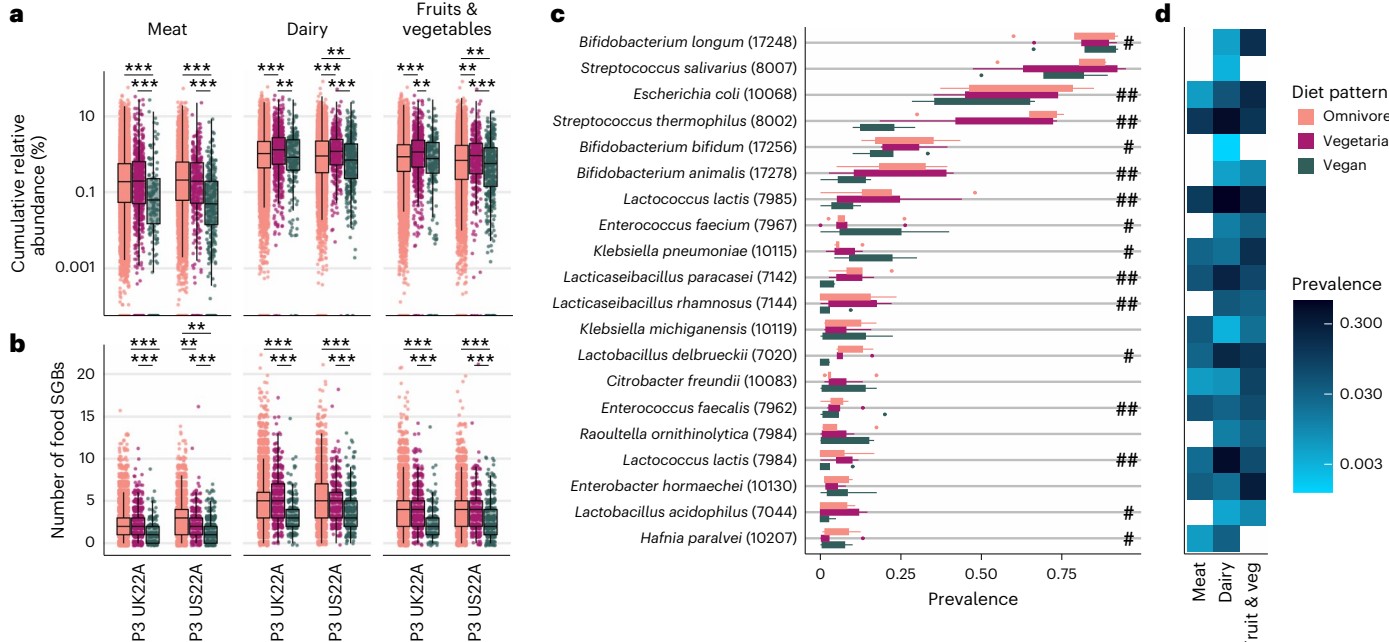

**Fig. 5 | Contribution of food microbes to the gut microbiome across diet patterns. a**, Cumulative relative abundance (log₁₀) and **b**, number of food SGBs (either meat, dairy, or fruits and vegetable-derived SGBs) within each individual's gut microbiome, coloured by diet pattern (omnivores: $n_{P3\,UK22A}$ = 11,533, $n_{P3\,US22A}$ = 7,228; vegetarians: $n_{P3\,UK22A}$ = 623, $n_{P3\,US22A}$ = 330; vegans: $n_{P3\,UK22A}$ = 197, $n_{P3\,US22A}$ = 373) and grouped by cohort (either P3 UK22A or P3 US22A; for all cohorts, see Extended Data Fig. 2). Asterisks denote significance level of BH-corrected Dunn's tests; **$P \le 0.01$, ***$P \le 0.001$ (Supplementary Tables 18 and 19). Boxplot parameters the same as in Fig. 1b. **c**, Prevalence of the 20 most common food SGBs (with their respective SGB IDs in parentheses) per diet pattern across all $n$ = 5 cohorts. Hashtags denote the number of cohorts (out of the three that were tested: P1, P3 UK22A, P3 US22A; Methods) in which two-sided chi-squared tests showed significant differences in SGB prevalence across all three diet patterns (Supplementary Table 17). **d**, Prevalence (log₁₀) of the 20 most common food SGBs across three major food categories (meat, dairy, fruits and vegetables) to indicate which food group each SGB is likely a signature of. White/blank boxes indicate that SGB was not prevalent in that particular food category.

This demonstrates the role that other foods may play in influencing this analysis based on entire FFQs versus highlighting only major food groups of interest. When comparing vegetarian with vegan gut microbiomes, the signature vegetarian microbes with the highest predictability are those linked with dairy consumption, for example, *S. thermophilus* (AUC = 0.72), *L. rhamnosus* (0.66), *L. delbrueckii* (0.70), *L. paracasei* (0.62), *L. lactis* (0.65) and *L. acidophilus* (0.68; Fig. 4f), aligning with our results thus far. These AUCs show that there exists a non-random, albeit mild, link between ingested food and the presence of specific species, suggesting causal links and potential transfer of microbes from food to gut.

### Diet-dependent gut microbiome contribution of food microbes

Until now, our results suggest the potential for diet patterns to select for gut microbes, but gut microbes might be derived directly from food itself[32], as may be the case for *S. thermophilus*, a common dairy component[27,33] that we found to be one of the most differentiating SGBs between diet patterns that differ in dairy consumption (Figs. 3h and 4b). To establish how many SGBs in each diet pattern's gut microbiome may be derived from food, we searched for food SGBs collated in 'curatedFoodMetagenomicData' (cFMD)[32] across our five cohorts and found 260 to be present (Methods). We found that the number of distinct food SGBs differed according to diet pattern, with significantly fewer food SGBs in vegan (zero-inflated negative binomial mixed model, $\beta$ = −0.36, $P < 0.001$; Fig. 5b and Methods) compared with omnivore and vegetarian ($\beta$ = 0.35, $P < 0.001$) microbiomes, but not between vegetarians and omnivores ($\beta$ = −0.004, $P < 0.686$).

When labelling these food SGBs as signatures of meat, dairy, and/ or fruits and vegetables if they had a prevalence >0.1% across these food groups, the effect of food group was significant and larger than the effect of diet pattern, with a greater number of food SGBs found for dairy ($\beta$ = 0.73, $P < 0.001$), followed by fruits and vegetables ($\beta$ = 0.47, $P < 0.001$). Thus, the largest factor impacting food-to-gut species sharing is the food group that SGBs are derived from, with greatest sharing from dairy products and lowest from meat. The number of food-associated SGBs was greatest in omnivores and vegetarians who both eat foods from the groups with the highest transmission (dairy, fruits and vegetables), and lowest in vegans who exclude meat and, more importantly here, dairy products, thus probably minimizing food-to-gut transmission rates.

We further found that the cumulative relative abundance of food SGBs in vegetarian gut microbiomes was significantly higher than in both omnivore (zero-inflated linear mixed-effects model, $\beta$ = 0.38, $P < 0.001$) and vegan ($\beta$ = 0.51, $P < 0.001$) gut microbiomes, and significantly lower in vegans compared with omnivores ($\beta$ = −0.12, $P = 0.026$; Fig. 5a). This again highlights the minimal food-to-gut species sharing in vegans. The fact that vegetarians had a greater cumulative relative abundance of food SGBs than omnivores but a similar number of distinct food SGBs may reflect the similar richness of food SGBs ingested (especially since meat-derived SGBs play an inferior role compared with SGBs derived from dairy and fruits and vegetables), but also the greater amount of fruits and vegetables ingested by vegetarians (Fig. 1c) instead of meat, which in turn drives a higher cumulative relative abundance of food SGBs. To summarize, dietary choices are linked to changes in the gut microbiome via not only potential selection but also food-to-gut acquisition.

### Food–gut shared microbes differ across diet patterns

We then identified 20 food SGBs with the highest prevalence across the five cohorts (Fig. 5c) and which major food groups these SGBs were signatures of (Fig. 5d). As expected, *S. thermophilus* was among

them, showing the greatest prevalence in dairy and significantly lowest prevalence in vegans (chi-squared tests; Methods, Fig. 5c and Supplementary Table 17). Similar patterns were observed for common dairy SGBs, for example, *L. acidophilus*, *L. delbrueckii*, *L. lactis*, *L. paracasei* and *L. rhamnosus*[27,28,33]—all SGBs that we found to be most greatly differentiated between vegan and non-vegan diet patterns. To lend more support to this hypothesis, we assessed omnivore and vegetarian frequency of dairy consumption (milk, yogurt, cheese, butter, other dairy) according to FFQs. We found that 96% of omnivores and 90% of vegetarians consume dairy at least once per week (Extended Data Fig. 3) with similar fractions (90% and 84% respectively) when restricting to fermented dairy products (yogurt and cheeses; Extended Data Fig. 3). Thus, we conclude that, while some microbe signatures of diets that include dairy could be selected to help digest dairy, others could be present in the gut microbiome as transient members derived from dairy foods themselves.

Several food SGBs with a high prevalence in vegans, such as *Enterobacter hormaechei*[34], *Citrobacter freundi*[35], *Raoultella ornithinolytica*[36] and *Klebsiella pneumoniae*[37,38] are members of the soil microbiome and/or nitrogen-fixing bacteria. Among them, *E. hormaechei* promotes growth in tomato and sweet pepper plants[34,39], while some strains of *K. pneumoniae* are nitrogen fixers and thus used as plant-growth promoters in wheat and soybeans[37,38]. This supports previous findings that, aside from more obvious possible sources of food-to-gut transmission such as cultured dairy products, agricultural practices could also play a role[40]. However, there is considerable phenotypic variation in these soil microbes, some of which may be opportunistic pathogens in humans and animals, hence their role in health still needs to be explored[36,41–43].

### Plant- and meat-specific microbial pathways and diet patterns

Since our results pointed towards gut microbial configurations seemingly adapted to the ingestion of major food groups, we explored this hypothesis by looking at the diet patterns' gut microbial functional potential (Methods). This revealed an array of plant-associated microbial pathways that were enriched in vegetarian and vegan diets compared with an omnivorous one. These included the conversion of simple carbohydrates (for example, D-galactose degradation pathway 6317) and of bioactive compounds (Extended Data Figs. 4–6 and Supplementary Tables 21–23). Among the latter, we identified the *myo-*, *chiro-* and *scillo*-inositol degradation pathway 7237, whose molecules (that is, the bioactive forms of inositol/vitamin $B_7$) represent the most abundant and accessible carbon and energy sources in plant-associated environments[44]. This pathway is particularly widespread across soil and rhizosphere bacteria and may provide a competitive advantage for growth and substrate utilization to microbes in plant-associated niches[44], such as those of a vegan or vegetarian gut microbiome. Also enriched in vegan and vegetarian gut microbes were pathways for the biosynthesis of chorismate (ARO and 6163), an intermediate in the production of various essential metabolites (for example, some aromatic amino acids, vitamins E and K, and ubiquinone[45]). These enzymatic routes are of note because they are shared among prokaryotes, including plant endosymbiotic cyanobacteria[45], as well as several eukaryotes, including ascomycete fungi[46]. This underscores yet again the enrichment of functions associated with plant ecological niches in vegan and vegetarian gut metagenomes.

In contrast, pathways overrepresented in omnivore gut microbiomes are involved in the breakdown of animal-derived foods and amino acid metabolism (Extended Data Figs. 4–6 and Supplementary Tables 20–22). These include the superpathway of L-threonine (THRESYN), and of L-serine and glycine biosynthesis (SER-GLYSYN), whose substrates are commonly found in red and white meat, and in dairy products[47,48]. Moreover, we found that omnivore microbiomes displayed the enzymatic machinery necessary for the salvage of essential cofactors that are abundant in foods of animal origin[49]. These cofactors included adenosylcobalamin (vitamin $B_{12}$) and folate

(vitamin $B_9$) from dietary precursors (for example, cobinamide in the COBALSYN pathway and 10-formyl-tetrahydrofolate in the 1CMET2 pathway, respectively[50,51]). The former is of particular note, since its precursors are derived from animal sources absent in vegan diets, making vitamin $B_{12}$ supplementation necessary for vegans[52]. In summary, gut microbial functional potential reveals diet-specific niches related to the metabolism of animal- or plant-derived foods, supporting the role of diet and its inclusion or exclusion of major food groups in shaping the gut microbial landscape both taxonomically and functionally.

## Discussion

Following diets that include or exclude major food groups such as meat, dairy, fruits and vegetables leaves its mark on the gut microbiome, which we characterized here by leveraging an integrated, multinational, metagenomic cohort of unprecedented size (21,561 individuals) with self-reported diet patterns. We found strong microbiome configurations for vegans, vegetarians and omnivores with several characteristic microbes that confirm and expand upon several previous findings. Among the 488 microbial signatures of an omnivore gut microbiome, we found species such as *A. putredinis*, *B. wadsworthia* and *R. torques*, that were generally linked to meat (especially red versus white meat) consumption. These species have been previously implicated in inflammatory diseases such as inflammatory bowel disease, colorectal cancer and an overall decrease in SCFAs, and were more likely to be associated with negative cardiometabolic health outcomes[19–21]. In contrast, signature microbes of a vegan gut microbiome, such as Lachnospiraceae, *Butyricicoccus* sp. and *R. hominis*, were linked to the consumption of fruits and vegetables, for example, due to their specialized role in fibre degradation, and are commonly described as producers of SCFAs[22–26]. These observations were also reflected by more signature vegan microbes associated with favourable cardiometabolic health than signature omnivore microbes and were paralleled by pathway-level microbiome characterization (Extended Data Figs. 4–6). Interestingly, we did not identify species in the *Segatella copri* (previously *Prevotella copri*) complex[53] as a strong signature of vegetarian or vegan diets (Supplementary Table 10), despite its hypothesized role in non-westernized populations characterized by fibre-rich diets[53].

Diets with high dairy components showed strong signatures of corresponding food microbes, in particular *S. thermophilus* and several lactic acid bacteria (for example, *L. acidophilus*, *L. paracasei* and *L. lactis*[27,28,33]), which are generally seen as health-associated gut microbial members. Vegan gut microbiomes had the highest prevalence of microbes shared with fruits and vegetables. In particular, we observed gut microbes that are shared with plant and soil microbiomes and have agricultural use in promoting plant growth through nitrogen fixation, such as *E. hormaechei* and some strains of *K. pneumoniae*[34,37–39]. These results are supported by previous findings[40] and provide evidence for an intriguing and yet-to-be-explored role of soil microbes in human, and in particular vegan, gut microbiomes.

Dietary factors within each diet pattern, such as the amount of healthy plant-based foods in one's diet, generally transcend the impact of overall diet patterns on the gut microbiome and are important for gut health. In particular, omnivores can modulate the fraction of gut microbial signatures shared with other diet patterns by adding plant-based food items in their diets (Fig. 1d,j). Since our data showed that omnivores on average ingest significantly fewer healthy plant-based foods than vegetarians or vegans, optimizing the quality of omnivore diets by increasing dietary plant diversity could lead to better gut health.

In summary, our work reinforces how humans can shape their own gut microbiomes, and by extension their health, directly through simple dietary choices as well as more indirectly through agricultural and food production practices. These diet pattern signatures will be important to inform experiments on specific interactions between single microbes (or genes) and food components, and are of potential use in a number of areas including improving (clinical) intervention studies

of different diet patterns and epidemiology studies where gut samples, but not detailed diet data, are available. Further research is still needed, for example at the strain level, to deduce food-to-gut transmission and explore microbes shared between human gut and food sources and to explore healthy practices within the three major diet patterns.

## Methods

Analyses were conducted using the R statistical language (v.4.2.2) unless otherwise stated.

### The ZOE PREDICT cohorts and two Italian datasets

This study encompassed two published, publicly available datasets (Tarallo et al. (2022)[12] with 118 individuals and De Filippis et al. (2019)[13] with 97 individuals) along with three ZOE PREDICT datasets: P1, a 90% UK/10% US cohort with 1,062 individuals; P3 UK22A, a UK cohort with 12,353 individuals; and P3 US22A, a US cohort with 7,931 individuals. Both P3 plus P2 clinical trials were registered at https://www.clinicaltrials.gov (clinical trial identifier for P3: NCT04735835; P2: NCT03983733) and ethics approval was obtained (P3 US protocol number (IRB): Pro00044316; P3 UK ethics review reference: HR-23/24-28300; P2 IRB: Pro00033432). Participants of P1, P2, P3 US22A and P3 UK22A all gave informed study consent either written or electronically. In addition, P3 US22A and P3 UK22A participants gave product research consent during the course of product purchase at ZOE Ltd. Only the US subset of P1 received modest direct financial compensation for their participation. All other participants did not receive direct financial compensation beyond reimbursement of expenses incurred. Individuals reported their dietary pattern (omnivore, vegetarian or vegan) and donated stool samples for shotgun metagenomic sequencing. In total, 656 vegans, 1,088 vegetarians and 19,817 omnivores were sampled. When possible, we also included samples from the ZOE PREDICT 2 (P2) cohort, which encompassed only omnivores from the United States (843 individuals), thus limiting its usability in this analysis. For this reason, most analyses presented here do not include P2 unless explicitly stated and any mention of 'the five cohorts' refers to all cohorts except P2. Detailed habitual dietary information from participants in all four ZOE PREDICT cohorts was obtained from quantitative food frequency questionnaires.

To further support the FFQs, we tested the Metagenomic Estimation of Dietary Intake (MEDI) tool[14], which uses food DNA in gut metagenomes to estimate and quantify food consumption (https://github.com/Gibbons-Lab/medi). We assessed how a MEDI-based classification of diet patterns would perform versus an FFQ-based classification, using what participants self-reported as the ground truth (only participants from P1, P3 22UKA and P3 22USA were considered, since these are the only cohorts with FFQs and all three diet patterns). To do so, we classified any sample in which MEDI found animal DNA as a non-vegan sample and any sample in which no animal DNA was found as a vegan sample. Similarly, we classified any sample whose FFQ reported the consumption of any animal product as a non-vegan sample and vice versa. Indeed, we found a lower prevalence of animal DNA among vegans vs non-vegans using the MEDI classification (Extended Data Fig. 7 and Supplementary Table 23), which highlights this tool's potential application in studies lacking data on overall dietary patterns. Compared with FFQs, however, we found MEDI unable to perform similarly well in predicting participants' diet patterns (chi-squared test, $P < 0.001$). While accurate thresholding of MEDI-derived statistics could improve performance, a deeper sequencing depth may be needed to substantially increase the tool's reliability by capturing a greater amount of food DNA, which is generally sparse in faecal samples. Since FFQs remain the gold standard and given the focus of our work on long versus short-term dietary patterns, we opted to base any analyses using food consumption data on FFQs, which have been extensively validated over time and in publications[10,11], and refer researchers to adopt a MEDI-based approach for studies lacking proper FFQs or as an additional validation tool.

FFQs were used to calculate hPDI values. Eighteen food groups were derived from and combined on the basis of the FFQ, segregated into quintiles and assigned positive or reverse scores[54]. A score of 5 was given to participants with an intake exceeding the largest positive score quintile and a score of 1 was given to those below the smallest quintile. Reverse scores received a reverse value. A final score was derived by summarizing the scores of each participant. Healthy plant-based foods received positive scores, whereas less healthy or unhealthy plant-based and animal-based foods received a reverse score. An hPDI value was able to be calculated for 900 individuals from P1 (841 omnivores, 49 vegetarians and 10 vegans), 12,091 from P3 UK22A (11,289 omnivores, 610 vegetarians and 192 vegans) and 7,375 from P3 US22A (6,720 omnivores, 309 vegetarians and 346 vegans). To compare hPDI values between the three diet patterns within each of the three PREDICT cohorts, we fit an ANOVA model to explain hPDI with diet pattern, sex, age (scaled) and BMI (scaled). This was followed by multiple comparisons using Tukey contrasts with a sandwich estimator to provide a heteroskedasticity-consistent estimator.

### DNA extraction, amplification and sequencing

P1 samples were extracted and sequenced as previously described and published[10]. P2 sample extraction and sequencing followed a similar protocol: samples were stored in Zymo buffer until DNA extraction at QIAGEN Genomic Services using DNeasy 96 PowerSoil Pro. P2 samples were sequenced on the Illumina NovaSeq 6000 platform using the S4 flow cell, targeting 7.5 Gbp per sample. Similarly, samples from both P3 cohorts were also stored in Zymo buffer until DNA extraction at Zymo using ZymoBIOMICS-96 MagBead DNA kit. P3 samples were sequenced on the Illumina NovaSeq 6000 platform using the S4 flow cell, targeting 3.75 Gbp per sample.

### Metagenome preprocessing and taxonomic profiling

We profiled all microbiome samples using MetaPhlAn 4 (v.4.beta.2, database v.Jan21_CHOCOPhlAnSGB_202103, with default parameters) to compute microbial relative abundances (Supplementary Code 1). These microbes were organized into SGBs that represent not only known species (for which reference genomes exist), but also unknown species currently described only by metagenome-assembled genomes (MAGs), thus expanding the resolution of taxonomic profiling[55,56].

### Alpha diversity

We calculated observed richness using MetaPhlAn's 'calculate_diversity' script (v.4.0.0)[55]. We considered all observations outside the 95% confidence intervals (CIs) to be outliers, which removed 22 samples. We tested for significant differences in alpha diversity between diet patterns within each of the five cohorts separately using Kruskal–Wallis rank sum tests with a significance level of 0.05, followed by Dunn's tests for multiple comparisons with $P$ values adjusted using the Benjamini–Hochberg (BH) method. In addition and to use a complementary, yet alternative approach, we fit a linear mixed model to predict observed richness with diet pattern, sex, age (scaled) and BMI (scaled) using the 'nlme' package (v.3.1.162). The model included cohort as a random effect. The model's intercept corresponded to diet pattern = omnivore, BMI = 0, age = 0 and sex = female. Confidence intervals (95%) and $P$ values were computed using a Wald $t$-distribution approximation (Supplementary Code 1). Observed richness differed significantly according to diet patterns, with a lower observed richness in vegans and vegetarians compared with omnivores (intercept corresponding to omnivores at 271.30, 95% CI [245.57, 297.04], $t(21,549) = 20.66, P < 0.001$; $\beta_{vegetarian} = -21.09$, 95% CI [−25.26, −16.93], $t(21,549) = -9.93, P < 0.001$; $\beta_{vegan} = -21.20$, 95% CI [−26.58, −15.82], $t(21,549) = -7.73, P < 0.001$; marginal $R^2 = 0.08$; Fig. 1b and Supplementary Table 7). To also compare vegetarians with vegans, we then built the same model with vegans in the intercept instead of omnivores. Thus, the model's intercept corresponded to diet pattern = vegan, BMI = 0, age = 0 and sex = female.

All other parameters remained the same. Using this model, there were no significant differences in observed richness between vegans and vegetarians (intercept corresponding to vegan at 250.10, 95% CI [223.99, 276.21], $t(21,549) = 18.77$, $P < 0.001$; $\beta_{vegetarian} = 0.11$, 95% CI [−6.47, 6.68], $t(21,549) = 0.03$, $P = 0.975$; Fig. 1b).

## Beta diversity
We calculated weighted and unweighted UniFrac, Aitchison and Bray-Curtis distances using MetaPhlAn's 'calculate_diversity' script (v.4.0.0)[55]. We tested for differences in beta diversity using the 'vegan' package (v.2.6.4) to run one PERMANOVA per cohort and per distance matrix with sex, age (scaled), BMI (scaled) and diet pattern (in that order) as explanatory variables and using adonis2 default settings (999 permutations, terms assessed sequentially). Beta diversity was plotted using principal coordinates analysis (PCoA) generated using the 'ape' package (v.5.7; Supplementary Code 1).

## Machine learning approaches
To link the participants' diet patterns to their microbiome community structure across P1, P3 UK22A, P3 US22A, De Filippis et al. (2019)[13] and Tarallo et al. (2022)[12] cohorts, we employed a machine learning approach: metAML[17] based on the random forest (RF) classification algorithm (scikit-learn Python library, v.0.22.2). The algorithm used was based on 1,000 estimator trees, 10 samples per leaf, no maximum depth, Gini impurity criterion and 10% of the total features' number in each tree. We performed two types of validation: (1) leave-one-dataset-out (LODO), which consisted of a training which encompasses all cohorts but one and was tested on the left-out cohort (and iteratively done for all cohorts); and (2) a hybrid approach named cross-LODO, which corresponded to a per-cohort (10-times, 10-fold) cross-validation, in which the rest of the cohorts were added to each training set as a support (Supplementary Code 1). First, we ran LODO and cross-LODO RF models on diet pattern pairs, using microbial SGB relative abundances as features. We also performed a set of experiments running LODO on SGB relative abundances as features together with sex, age and BMI to test the microbiome's ability in distinguishing dietary habits when accounting for human interpersonal variability (Supplementary Table 9 and Supplementary Code 1). To test for potential data leakage and overfitting, we randomly swapped the diet pattern labels in a cross-LODO experiment, which, as expected, did not result in AUCs above 0.51.

Moreover, we performed per-cohort cross-validations on the cohorts P1, P2, P3 US22A and P3 UK22A to predict the presence of those SGBs found at a prevalence between 10% and 90%, using the participants' dietary composition as features estimated by their FFQs. The average predictability of each SGB's presence based on FFQs was then computed by meta-analysing the four AUCs (see following section). Final AUCs in each case were computed as an average over 100 tests (cross-validation and cross-LODO) and an average over 10 tests (LODO). Cross-LODO ROC curves were plotted as a linear interpolation (scipy v.1.11.4) over the 100 tests, with 95% confidence intervals computed on the basis of the bootstrap standard error under the assumption of a $t$-distribution.

## Meta-analysis approaches
Several statistical association measures were computed on each cohort separately and then pooled via inverse-variance weighting (meta-analysis) of the relevant coefficients. To determine differentially abundant microbes between the diet patterns, for each of the five main cohorts analysed in this study, we built linear models to assess the differentially abundant SGBs between diet pairs ('omnivore vs vegetarian', 'omnivore vs vegan' and 'vegetarian vs vegan'). Linear models were fit to each diet pair (as a categorical variable) on the arcsin-square-root-transformed SGB's relative abundance to compensate for the proportions variance instability, and were adjusted by sex, age and BMI. The corresponding mean abundance difference in the two diets was transformed into a standardized mean difference (adjusted Cohen's $d$). The pooled estimate of effect sizes from linear models implements a random-effects meta-analysis with DerSimonian-Laird heterogeneity (Supplementary Code 1 and Code availability statement). In an additional meta-analysis, the cross-validation AUCs resulting from the RF done on FFQs to predict the presence/absence of SGBs (see section above) were meta-analysed over all the ZOE PREDICT cohorts, including P2, using the same Python script just described with the AUC standard error computed by metAML.

To determine the links between major food groups and SGB relative abundance, the partial Spearman's correlation (adjusted by sex, age and BMI) between each SGB's relative abundance and the individual intake of meat (white and red), dairy products, fruits and vegetables, was computed after having summed the total intake of single FFQ items belonging to each group, for all the ZOE PREDICT cohorts, including P2. Correlations were calculated using individuals from diets that consumed that particular food group; for example, correlations between SGB abundances and meat were calculated using only omnivores, while correlations concerning fruits and vegetables were calculated using individuals from all three diet patterns. An additional meta-analysis was conducted on the partial Spearman's correlation between each SGB's relative abundances and hPDI. In these cases, the pooled estimates of correlations were computed using the meta package in R (v.7.0-0) and were based on the standard error of the Fisher $Z$ correlation transformation. For all meta-analyses, Wald test $P$ values and correlation $P$ values for all SGBs evaluated were adjusted for false discovery rate (BH) in each cohort separately and in each meta-analysis. Statistical significance was defined as a $q$-value < 0.1 (Supplementary Code 1).

## ZOE MB Health ranks
The ZOE Microbiome Ranking 2024 ranks SGBs that significantly correlate with a set of cardiometabolic markers such as BMI, blood pressure and lipoproteins, and was defined on the basis of five ZOE PREDICT studies (P1, P2, P3 US21, P3 UK22A and P3 US22A) and ~35,000 individuals[30]. We searched for these ranked SGBs across our multicohort dataset and compared mean ranks of SGB signatures of the various diet patterns (as determined by the meta-analysis described above) by conducting two-sample $t$-tests with equal variance (determined using Levene's test, $P > 0.05$ for all diet pattern pairs) on the ranks for the statistically significant differentially abundant SGBs between each diet pattern pair.

## Food microbiome
To determine how many and which SGBs in each diet pattern's gut microbiome may be derived from food, we used the cFMD database of metagenomes sampled from various food sources, which defined food SGBs as those found with a relative abundance ≥0.1% in ≥4 food samples from taxonomic profiles[32]. This resulted in 816 food SGBs, which we then searched for across our five cohorts. We found 263 of these 816 food SGBs to be present across our whole dataset. We considered these food SGBs to be signatures of meat, dairy, and/or fruits and vegetables if they had a prevalence of >0.1% across food samples belonging to the three aforementioned food groups. To establish whether the number of food SGBs present in gut microbiomes differs according to diet pattern or food group they are a signature of (meat, dairy, or fruits and vegetables), we fit a zero-inflated negative binomial mixed model estimated by maximum likelihood from the 'NBZIMM' package (v.1.0) to predict the number of food SGBs per sample with diet pattern, food group, sex, age and BMI. The model included cohort as a random effect with conditional $R^2 = 0.10$, marginal $R^2 = 0.08$, and intercept corresponding to omnivore microbiomes and meat SGBs at 0.77, $P < 0.001$. To also compare vegetarians with vegans, we then built the same model with vegans in the intercept instead of omnivores. All other parameters remained the same. Similarly, to establish

whether the cumulative relative abundance of food SGBs in gut microbiomes differs according to diet pattern or food group, we fit another zero-inflated linear mixed model estimated by maximum likelihood with the same model structure as above. The model's intercept corresponded to omnivore microbiomes and meat SGBs at −0.41, $P < 0.001$, with conditional $R^2 = 0.05$ and marginal $R^2 = 0.05$. Again, vegans were moved to the intercept in a following model with the same parameters. In addition and using a slightly different approach, we tested for differences in the number of food SGBs and in their cumulative relative abundance between the diet patterns and within each food group and within each cohort using Dunn's tests coupled with a BH correction for multiple testing.

To establish whether there were any significant differences in the prevalence of the 20 most common food SGBs between the three diet patterns, we ran a chi-squared test on the number of omnivore, vegetarian and vegan microbiomes in which each of the SGBs was present versus absent with the option to compute $P$ values by Monte Carlo simulation (99,999 replicates). The tests were run for the larger cohorts separately, namely, for P1, P3 UK22A and P3 US22A.

### Gut microbial functional potential

To generate gut microbial functional potential, we ran HUMAnN (v.3.6)[57] with default parameters (Supplementary Code 1). We focused on the pathway abundance output and removed any unmapped or unintegrated pathways, as well as pathways with a prevalence of <0.05 across samples of at least one diet pattern and a coverage of <0.2. This left us with 87 pathways in P1, 85 pathways in P3 UK22A and 87 pathways in P3 US22A. We then measured the statistical association between the relative abundances of these pathways and each diet pattern pair, which we first computed on each of the three PREDICT cohorts separately and then meta-analysed as described in detail above (Supplementary Code 1).

### Reporting summary

Further information on research design is available in the Nature Portfolio Reporting Summary linked to this article.

### Data availability

The publicly available datasets used in this work are available from their respective publications in refs. 12,13. Raw metagenomic samples are provided for all participants of the ZOE PREDICT studies. Specifically, PREDICT 1 has already been made publicly available as reported previously[10] under the NCBI-SRA bioproject ID PRJEB39223, whereas PREDICT 2 is deposited in EBI under accession number PRJEB75460, and PREDICT 3 cohorts under EBI accession numbers PRJEB75463 and PRJEB75464. Sex, age, BMI, country and the quantitative taxonomic profiles are available for each sample within the curatedMetagenomicData package[58]. The ZOE Microbiome Rankings for the full list of species are made available (and kept up-to-date) at https://zoe.com/our-science/microbiome-ranking. ZOE is the owner of the pseudonymized data and metadata and researchers interested in follow-up studies requiring additional specific metadata information should fill out a research request proposal at https://zoe.com/our-science/collaborate that will be evaluated by a subpanel of the ZOE Scientific Advisory Board once per month for their priority, relevance and in compliance with privacy and data protection regulations.

### Code availability

The code for the analyses conducted here is provided in Supplementary Code 1. The pooled estimate of effect sizes from linear models was computed on the basis of the pipeline in GitHub at https://github.com/waldronlab/curatedMetagenomicDataAnalyses/blob/main/python_tools/metaanalyze.py. An importable meta-analysis Python library is also freely available in GitHub at https://github.com/SegataLab/inverse_var_weight/blob/main/meta_analyses.py.

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

## Acknowledgements

We thank all the participants of the ZOE PREDICT studies, as well as members of the Computational Metagenomics Lab for their continued input and support, and members of ZOE Ltd. that made this research possible. This work was supported by Zoe Ltd. and co-funded by the European Union under Horizon Europe Programme CoDiet [101084642] to N.S. with UK activities supported by UK Research and Innovation under the UK government's Horizon Europe funding guarantee. More information on the CoDiet project can be found at https://www.codiet.eu/. This work was also partially supported by the European Research Council (ERC-STG project MetaPG-716575 and ERC-CoG microTOUCH-101045015) to N.S., by the European Union's Horizon 2020 programme (ONCOBIOME-825410 project, MASTER-818368 project and IHMCSA-964590 project) to N.S., by

the MUR PNRR project INEST-Interconnected Nord-Est Innovation Ecosystem (ECS00000043) funded by the NextGenerationEU to N.S., by the National Cancer Institute of the National Institutes of Health (1U01CA230551 to N.S.) and by the Premio Internazionale Lombardia e Ricerca 2019 to N.S. G.F. was funded by the European Union under the Marie Sklodowska-Curie grant agreement no. 101152592–plasticOME. Views and opinions expressed are, however, those of the author(s) only and do not necessarily reflect those of the European Union or European Climate, Infrastructure and Environment Executive Agency (CINEA). Neither the European Union nor the granting authority can be held responsible for them.

## Author contributions

G.F., P.M., J.W., F.A., T.D.S. and N.S. conceived and designed the study. Metadata processing was performed by F.A., A.A., E.B., A.C.C., L.F., J.C.P., R.D., K.M.B. and S.E.B. Metagenomic data were pre-processed and quality controlled by F.A. and P.M. G.F. and P.M. performed the analyses. N.C., V.H., E.P., G.P. and R.D. contributed to the analyses. Data were visualized by G.F. and P.M. L.R. and V.H. contributed to the interpretation of the results. G.F., P.M. and N.S. wrote the paper with contribution and editing from all authors.

## Competing interests

T.D.S. and J.W. are co-founders of ZOE Ltd. F.A., S.E.B., T.D.S. and N.S. are consultants to ZOE Ltd. A.A., E.B., A.C.C, L.F., J.C.P., R.D., J.W. and K.M.B. are or have been employees of Zoe Ltd. A.A., J.C.P., R.D., J.W., A.C.C, K.M.B., S.E.B., T.D.S., F.A. and N.S. receive options in ZOE Ltd. All other authors declare no competing interests.

## Additional information

**Extended data** is available for this paper at https://doi.org/10.1038/s41564-024-01870-z.

**Correspondence and requests for materials** should be addressed to Nicola Segata.

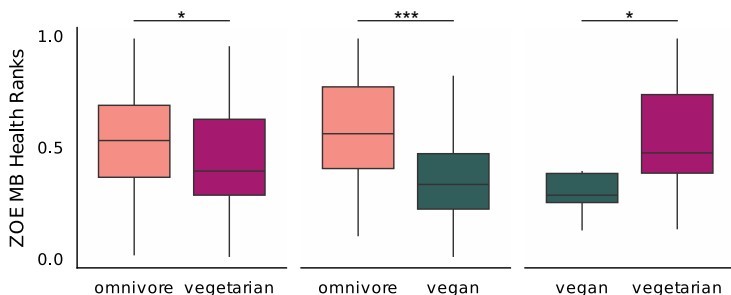

**Extended Data Fig. 1 | Distribution of ZOE CMH rankings of SGBs signature of the three diet patterns.** ZOE MB Health Ranks (y-axis; ranks closer to 0 indicate more favorable and those closer to 1 indicate less favorable cardiometabolic health outcomes) of all SGBs statistically significantly differentially abundant between each diet pattern pair (x-axis; $n_{omnivore-vegetarian}$ = 600 SGBs, $n_{omnivore-vegan}$ = 724, $n_{vegan-vegetarian}$ = 41), colored by diet pattern (pink = omnivore, purple = vegetarian, green = vegan). Boxplots show the median, 25th and 75th percentiles, and whiskers extend to 1.5 times the interquartile range. Asterisks denote significance level of two sample t-tests (**Methods**), with * $0.05 > p < 0.01$ and *** $p \le 0.001$.

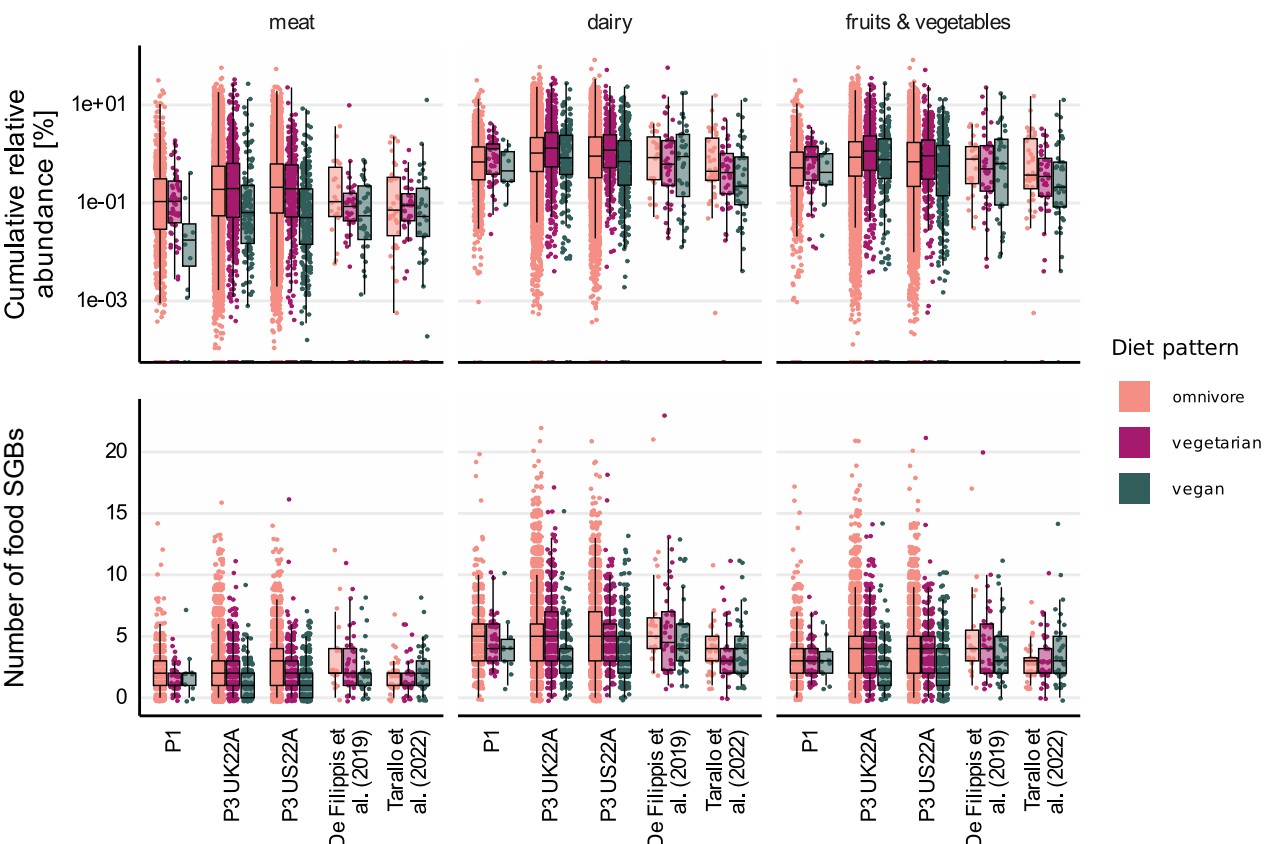

**Extended Data Fig. 2 | Microbial food-to-gut transmission across the diet types and major food categories.** Cumulative relative abundance (in %, $\log_{10}$ scale; upper panels) and prevalence, that is, count, (lower panels) of food SGBs (either meat, dairy, or fruits and vegetable-derived SGBs) within each individual, colored by diet pattern (pink = omnivore, purple = vegetarian, green = vegan) and grouped by cohort (P1, P3 UK22A, P3 US22A, De Filippis et al. [13], and

Tarallo et al. [12]; Supplementary Tables 19, 20). Boxplots show the median, 25th and 75th percentiles, and whiskers extend to 1.5 times the interquartile range. Omnivores: $n_{P1} = 991$, $n_{P3\,UK22A} = 11,533$, $n_{P3\,US22A} = 7,228$, $n_{De\,Filippis} = 23$, $n_{Tarallo} = 40$; vegetarians: $n_{P1} = 59$, $n_{P3\,UK22A} = 623$, $n_{P3\,US22A} = 330$, $n_{De\,Filippis} = 38$, $n_{Tarallo} = 38$; vegans: $n_{P1} = 10$, $n_{P3\,UK22A} = 197$, $n_{P3\,US22A} = 373$, $n_{De\,Filippis} = 36$, $n_{Tarallo} = 40$ individuals.

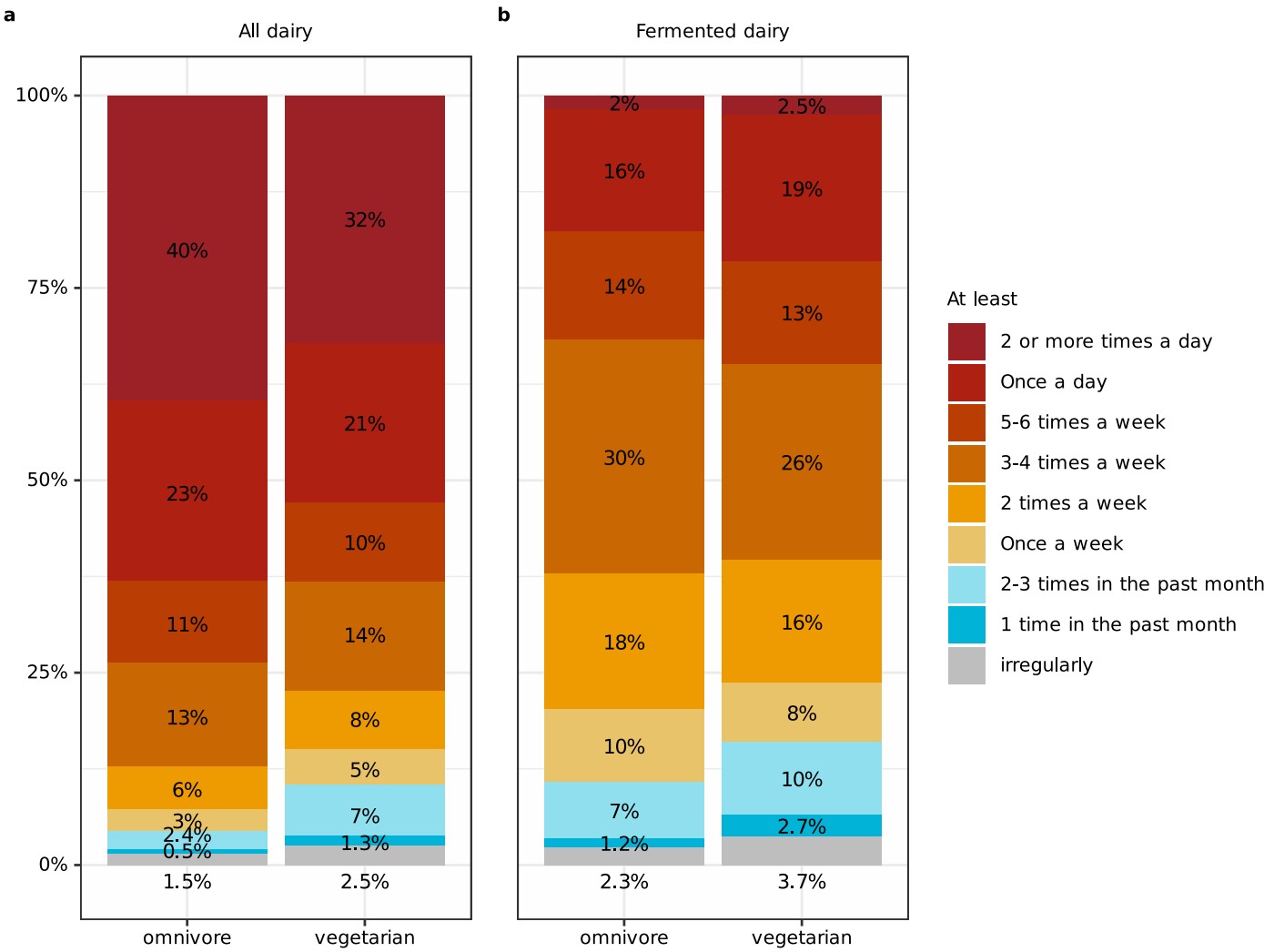

**Extended Data Fig. 3 | Frequency of dairy consumption across omnivores and vegetarians in P3 UK22A and P3 US22A according to FFQs. a** Percentage (y-axis) of omnivores and vegetarians (x-axis) that consume dairy products (milk, yogurt, cheese, butter, or other dairy) between 'two or more times per day' to 'irregularly'. The consumption frequency categories were given by the FFQs. Percentages within the bar plots indicate the dairy consumption prevalence of that diet pattern in that consumption category. **b** Same as in a, but considering only fermented dairy products (yogurt and cheeses).

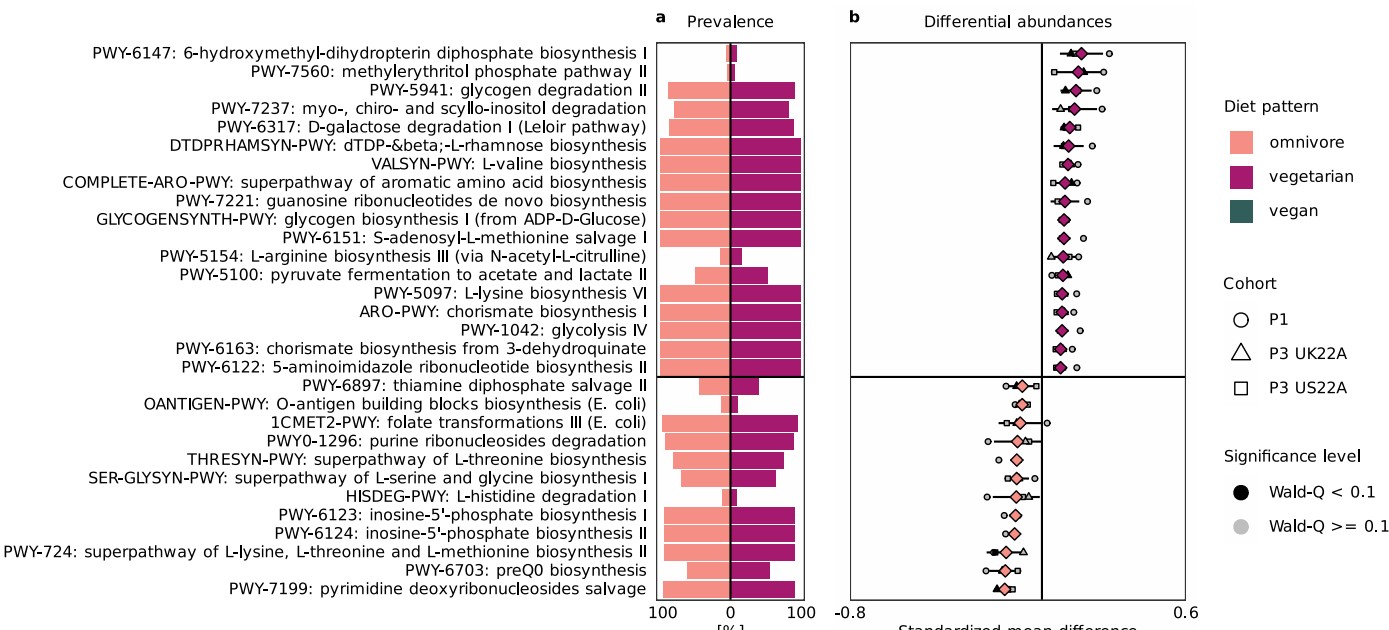

**Extended Data Fig. 4 | Potential microbial functional signatures of omnivore and vegetarian gut microbiomes. a** Prevalence (in %) of each functional pathway (y-axis) in omnivore (pink, left bars) and vegetarian (purple, right bars) gut microbiomes. **b** Meta-analyzed correlations between pathway relative abundance and diet pattern (omnivore vs vegetarian) for the top 30 pathways with the largest absolute standardized mean difference, upper and lower confidence intervals. Purple dots to the right indicate pathway-associations with vegetarians, while pink dots to the left indicate pathway-associations with omnivores. Also shown in smaller shapes are the per-cohort correlations, with shapes filled in black indicating a Wald q-value < 0.1 and those filled in gray indicating a Wald q-value ≥ 0.1. The black horizontal bar indicates the separation between the correlations with omnivores vs vegetarians for ease of visualization only. Shown are only the pathways with a prevalence of less than 0.05 across samples of at least one diet pattern, a coverage less than 0.2, and which were significant at q < 0.1. $n_{omnivores} = 19,817$, $n_{vegetarians} = 1,088$.

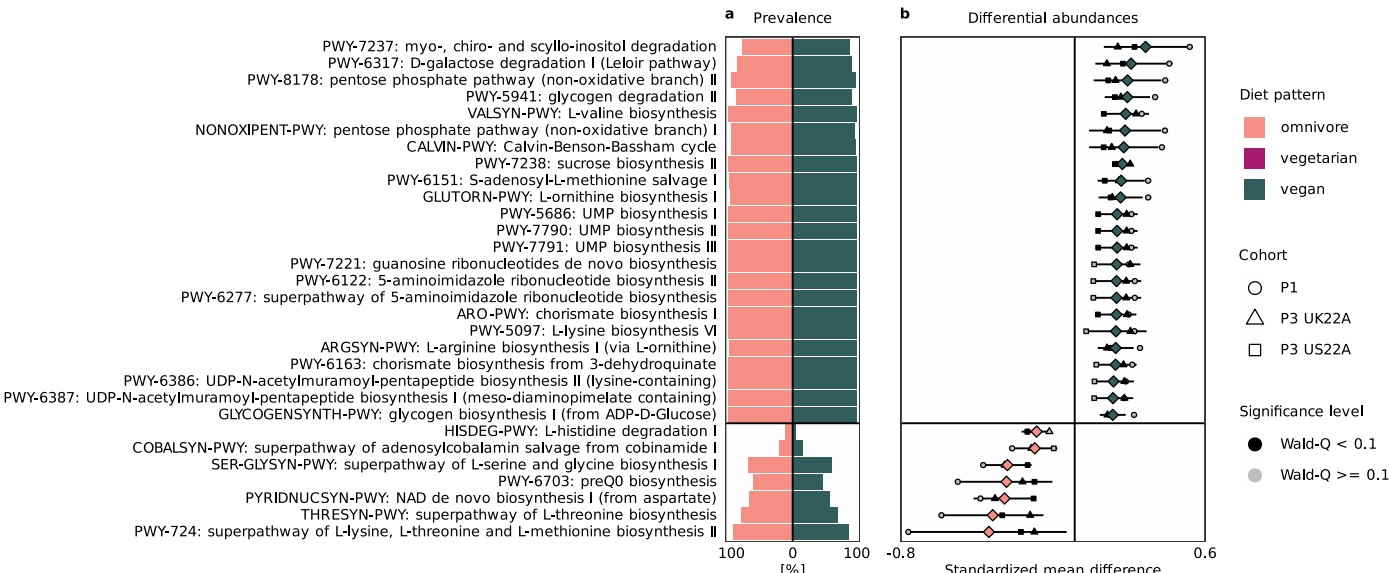

**Extended Data Fig. 5 | Potential microbial functional signatures of omnivore and vegan gut microbiomes. a** Prevalence (in %) of each functional pathway (y-axis) in omnivore (pink, left bars) and vegan (green, right bars) gut microbiomes. **b** Meta-analyzed correlations between pathway relative abundance and diet pattern (omnivore vs vegan) for the top 30 pathways with the largest absolute standardized mean difference, upper and lower confidence intervals. Green dots to the right indicate pathway-associations with vegans, while pink dots to the left indicate pathway-associations with omnivores.

Also shown in smaller shapes are the per-cohort correlations, with shapes filled in black indicating a Wald q-value < 0.1 and those filled in gray indicating a Wald q-value ≥ 0.1. The black horizontal bar indicates the separation between the correlations with omnivores vs vegans for ease of visualization only. Shown are only the pathways with a prevalence of less than 0.05 across samples of at least one diet pattern, a coverage less than 0.2, and which were significant at q < 0.1. $n_{omnivores} = 19,817$, $n_{vegans} = 656$.

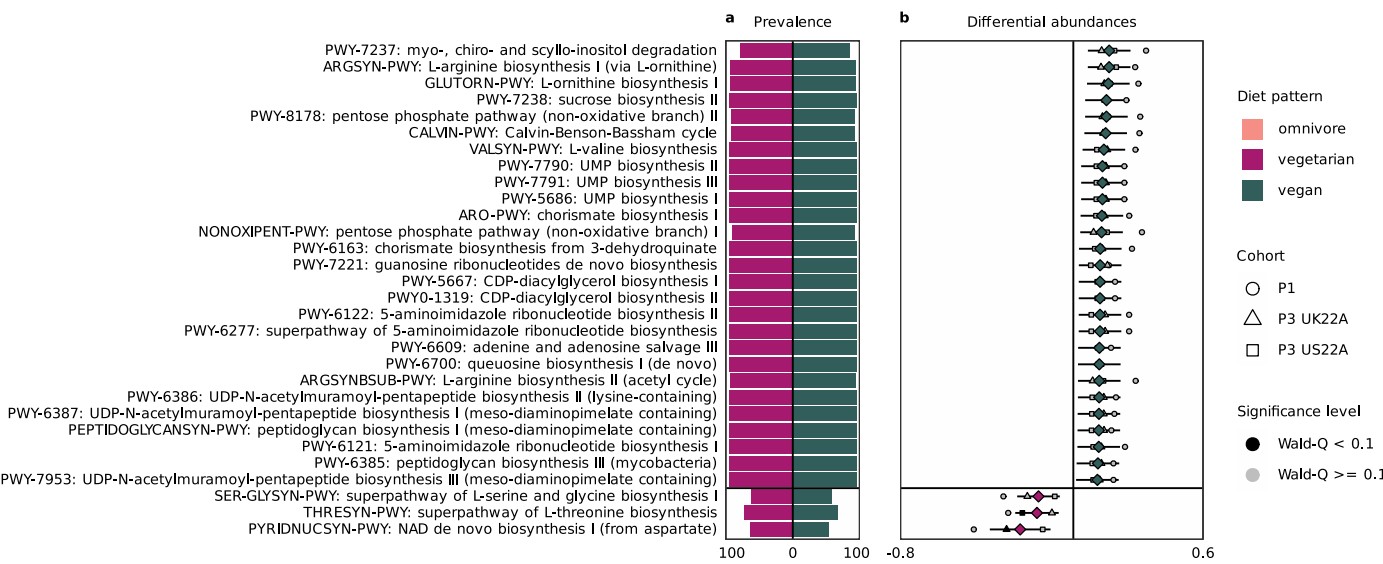

**Extended Data Fig. 6 | Potential microbial functional signatures of vegetarian and vegan gut microbiomes. a** Prevalence (in %) of each functional pathway (y-axis) in vegetarian diet (purple, left bars) and vegan (green, right bars) gut microbiomes. **b** Meta-analyzed correlations between pathway relative abundance and diet pattern (vegetarian vs vegan) for the top 30 pathways with the largest absolute standardized mean difference, upper and lower confidence intervals. Green dots to the right indicate pathway-associations with vegans, while purple dots to the left indicate pathway-associations with vegetarians.

Also shown in smaller shapes are the per-cohort correlations, with shapes filled in black indicating a Wald q-value < 0.1 and those filled in gray indicating a Wald q-value ≥ 0.1. The black horizontal bar indicates the separation between the correlations with vegetarians vs vegans for ease of visualization only. Shown are only the pathways with a prevalence of less than 0.05 across samples of at least one diet pattern, a coverage less than 0.2, and which were significant at q < 0.1. $n_{vegetarians} = 1,088$, $n_{vegans} = 656$.

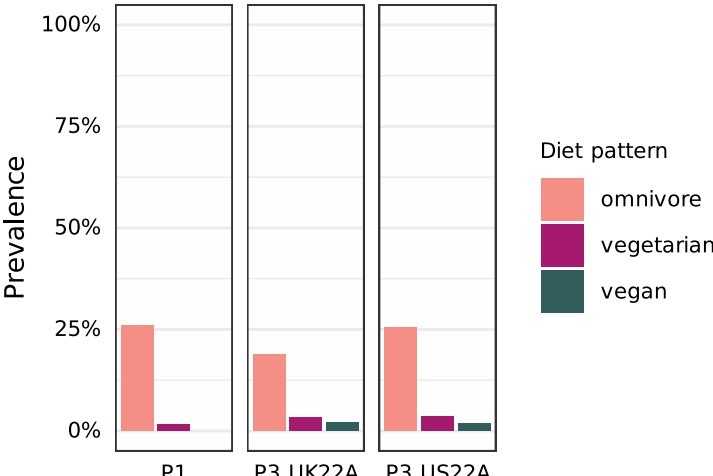

**Extended Data Fig. 7 | Estimation of animal-based food consumption based on metagenomic reads.** Prevalence (in %) of animal DNA as estimated by MEDI using the gut metagenomes of omnivores (bars in pink), vegetarians (bars in purple), and vegans (bars in green) in the P1, P3 UK22A and P3 US22A cohorts (Supplementary Table 24).

# Reporting Summary

## Statistics

For all statistical analyses, confirm that the following items are present in the figure legend, table legend, main text, or Methods section.

| n/a | Confirmed | |
|---|---|---|
| ☐ | ☒ | The exact sample size (*n*) for each experimental group/condition, given as a discrete number and unit of measurement |
| ☐ | ☒ | A statement on whether measurements were taken from distinct samples or whether the same sample was measured repeatedly |
| ☐ | ☒ | The statistical test(s) used AND whether they are one- or two-sided<br>*Only common tests should be described solely by name; describe more complex techniques in the Methods section.* |
| ☐ | ☒ | A description of all covariates tested |
| ☐ | ☒ | A description of any assumptions or corrections, such as tests of normality and adjustment for multiple comparisons |
| ☐ | ☒ | A full description of the statistical parameters including central tendency (e.g. means) or other basic estimates (e.g. regression coefficient) AND variation (e.g. standard deviation) or associated estimates of uncertainty (e.g. confidence intervals) |
| ☐ | ☒ | For null hypothesis testing, the test statistic (e.g. *F*, *t*, *r*) with confidence intervals, effect sizes, degrees of freedom and *P* value noted<br>*Give P values as exact values whenever suitable.* |
| ☒ | ☐ | For Bayesian analysis, information on the choice of priors and Markov chain Monte Carlo settings |
| ☒ | ☐ | For hierarchical and complex designs, identification of the appropriate level for tests and full reporting of outcomes |
| ☐ | ☒ | Estimates of effect sizes (e.g. Cohen's *d*, Pearson's *r*), indicating how they were calculated |

*Our web collection on <u>statistics for biologists</u> contains articles on many of the points above.*

## Software and code

Policy information about <u>availability of computer code</u>

| Data collection | n/a |
|---|---|
| Data analysis | R (version 4.2.2), MetaPhlAn 4 (version 4.beta.2, database vJan21_CHOCOPhlAnSGB_202103), HUMAnN (version 3.6), scikit-learn python library (version 0.22.2). The code for the analyses conducted here is provided in Supplementary File 1. The pooled estimate of effect sizes from linear models was computed based on the pipeline https://github.com/waldronlab/curatedMetagenomicDataAnalyses/blob/main/python_tools/metaanalyze.py. An importable meta-analysis python library is also freely available at https://github.com/SegataLab/inverse_var_weight/blob/main/meta_analyses.py. |

For manuscripts utilizing custom algorithms or software that are central to the research but not yet described in published literature, software must be made available to editors and reviewers. We strongly encourage code deposition in a community repository (e.g. GitHub). See the Nature Portfolio <u>guidelines for submitting code & software</u> for further information.

## Data

Policy information about availability of data

All manuscripts must include a data availability statement. This statement should provide the following information, where applicable:

- Accession codes, unique identifiers, or web links for publicly available datasets
- A description of any restrictions on data availability
- For clinical datasets or third party data, please ensure that the statement adheres to our policy

The publicly available datasets used in this work are available from their respective publications in Tarallo et al. 2022 and De Filippis et al. 2019. Raw metagenomic samples are provided for all participants of the ZOE PREDICT Studies. Specifically, PREDICT 1 has already been made publicly available as reported previously (Asnicar, Berry, et al. 2021) under the NCBI-SRA bioproject ID PRJEB39223, whereas the PREDICT 2 is deposited in EBI under accession number PRJEB75460, and PREDICT 3 cohorts under EBI accession numbers PRJEB75463 and PRJEB75464. Sex, age, BMI, country, and the quantitative taxonomic profiles are available for each sample within the curatedMetagenomicData package Pasolli et al. 2017. The ZOE Microbiome Rankings for the full list of species are made available (and kept up-to-date) at https://zoe.com/our-science/microbiome-ranking and in their current version are reported in Supplementary Table 14. ZOE is the owner of the pseudonymized data and metadata and researchers interested in follow-up studies requiring additional specific metadata information should fill out a research request proposal at https://zoe.com/our-science/collaborate [to appear upon acceptance] that will be evaluated by a sub-panel of the ZOE Scientific Advisory Board once per month for their priority, relevance and in compliance with privacy and data protection regulations.

## Research involving human participants, their data, or biological material

Policy information about studies with human participants or human data. See also policy information about sex, gender (identity/presentation), and sexual orientation and race, ethnicity and racism.

| Reporting on sex and gender | Data on sex (not gender) were collected with informed consent of participants (see Data Availability Statement). Sex was considered throughout the analysis as an explanatory variable, as is indicated throughout the Methods. |
|---|---|
| Reporting on race, ethnicity, or other socially relevant groupings | Data on socially relevant variables such as race or ethnicity were neither collected nor considered in this research, where the focus was on dietary patterns pertaining to veganism, vegetarianism and mixed diets. |
| Population characteristics | Participants were aged 52 +/- 12.5 years (mean +/- standard deviation). Genotype and diagnosis information was not collected. |
| Recruitment | Information pertaining to the publicly available datasets used in this work are available from their respective publications: (Tarallo et al. 2022) and (De Filippis et al. 2019). Participants of P1, P2, P3 US22A, and P3 UK22A all gave informed study consent either written or electronically. In addition, P3 US22A and P3 UK22A participants gave product research consent during the course of product purchase at ZOE Ltd. Only the US subset of P1 received modest direct financial compensation for their participation. All other participants did not receive direct financial compensation beyond reimbursement of expenses incurred. |
| Ethics oversight | Information pertaining to the publicly available datasets used in this work are available from their respective publications: Asnicar et al. 2021, Tarallo et al. 2022 and De Filippis et al. 2019. Both P3 plus P2 clinical trials were registered at https://www.clinicaltrials.gov (clinical trial identifier for P3: NCT04735835; P2: NCT03983733) and ethical approval was obtained (P3 US protocol number (IRB): Pro00044316; P3 UK ethical review reference: HR-23/24-28300; P2 IRB: Pro00033432). |

Note that full information on the approval of the study protocol must also be provided in the manuscript.

# Field-specific reporting

Please select the one below that is the best fit for your research. If you are not sure, read the appropriate sections before making your selection.

☒ Life sciences    ☐ Behavioural & social sciences    ☐ Ecological, evolutionary & environmental sciences

For a reference copy of the document with all sections, see nature.com/documents/nr-reporting-summary-flat.pdf

# Life sciences study design

All studies must disclose on these points even when the disclosure is negative.

| Sample size | This study encompassed two published, publicly available datasets (Tarallo et al. 2022 with 118 individuals and De Filippis et al. 2019 with 97 individuals) along with three ZOE PREDICT datasets: P1, a UK cohort with 1,062 individuals; P3 UK22A a UK cohort with 12,353 individuals; and P3 US22A, a US cohort with 7,931 individuals. In total, 656 vegans, 1,088 vegetarians, and 19,817 omnivores were sampled. When possible, we also included samples from the ZOE PREDICT 2 (P2) cohort, which encompassed only omnivores from the US (843 individuals), thus limiting its usability in this analysis. |
|---|---|
| Data exclusions | Data was only excluded in the alpha diversity analysis, in which we considered all observations outside the 95% CI to be outliers, which removed 22 out of the 21,561 samples. |
| Replication | The gut microbial signatures of the 3 diet patterns were replicable across 5 independent cohorts (P1, P3 US22A, P3 UK22A, De Filippis et al. |

| Replication | (2019) and Tarallo et al. (2022) using a cross-LODO machine learning approach. Linking these patterns to FFQs and individual physiological data was replicable across 4 cohorts for which FFQ data were present (P1, P2, P3 US22A, and P3 UK22A) using a meta-analytical approach. |
| Randomization | Participants were grouped into dietary patterns according to their reported dietary patterns. Randomization was only necessary when cross-LODO machine learning was applied, in which case a per-cohort (ten-times, ten-folds) cross-validation was performed, in which the rest of the cohorts is added to each training set as a support. |
| Blinding | Authors who extracted and sequenced stool samples did not conduct the microbiome analysis. |

# Reporting for specific materials, systems and methods

We require information from authors about some types of materials, experimental systems and methods used in many studies. Here, indicate whether each material, system or method listed is relevant to your study. If you are not sure if a list item applies to your research, read the appropriate section before selecting a response.

## Materials & experimental systems

| n/a | Involved in the study |
|-----|----------------------|
| ☒ ☐ | Antibodies |
| ☒ ☐ | Eukaryotic cell lines |
| ☒ ☐ | Palaeontology and archaeology |
| ☒ ☐ | Animals and other organisms |
| ☒ ☐ | Clinical data |
| ☒ ☐ | Dual use research of concern |
| ☒ ☐ | Plants |

## Methods

| n/a | Involved in the study |
|-----|----------------------|
| ☒ ☐ | ChIP-seq |
| ☒ ☐ | Flow cytometry |
| ☒ ☐ | MRI-based neuroimaging |

## Plants

| Seed stocks | n/a |
| Novel plant genotypes | n/a |
| Authentication | n/a |

