## [Peer Review File · Nature Microbiology]

Gut microbiome signatures of vegan, vegetarian and omnivore diets and associated health outcomes across 21,561 individuals

Corresponding Author: Professor Nicola Segata

Version 0:

Reviewer comments:

Reviewer #1

(Remarks to the Author)

Summary -

The authors present a large cross-sectional study (N = 21,561) of how gut metagenomic features (microbial taxonomic units) are associated with dietary intake patterns. General dietary patterns were known for all of the individuals (656 vegans, 1,088 vegetarians, and 19,817 mixed diets), and more specific food frequency data was available for a subset (~13,000). The composition of the microbiome could classify dietary type (AUC = 0.85) and several taxonomic units were associated with specific foods and with host phenotypes. Not only did the authors identify human gut taxa that appeared to respond to the inclusion of specific foods, but they also found food-associated taxa that appear to serve as biomarkers of eating specific foods (e.g., dairy-associated lactobacilli or soil-associated taxa found on vegetables). This is an interesting and useful resource, presented in a well-written manuscript. I commend the authors on a nice paper. My major concerns involve the lack of detail on analysis code, along with a few minor concerns, outlined below.

Major Comments -

The authors do not provide a link to their analysis code (e.g., a GitHub repository). I see that they are using biobakery tools, but they need to provide the actual commands/code that was run to generate their results. There's also reference to 'custom scripts', which are not provided.

The thrust of this paper is to look for bacterial taxonomic biomarkers of dietary intake, but there are a few existing approaches to dietary assessment directly from data that are not discussed by the authors. It would be useful to compare and contrast their results to these other methods. For example, prior work has proposed using metabolomic biomarkers in stool to assess intake (<https://www.ncbi.nlm.nih.gov/pmc/articles/PMC9021190/>). Amplicon sequencing of chloroplasts in human stool has also been proposed (<https://www.pnas.org/doi/10.1073/pnas.2304441120>), in addition to direct detection of whole-genome food-related DNA in metagenomes (<https://www.biorxiv.org/content/10.1101/2024.02.02.578701v1>).

The authors focus on taxonomic biomarkers of dietary intake, but they don't try to identify functional genes. Why not? You get both sets of info from your metagenomes. It may be that functional gene profiles are more informative than taxonomic profiles.

Minor Comments -

line 18: change 'positively correlate with negative host' to 'negatively correlated with host'

line 28: 'risk' -> 'risk factor'

line 30-31: 'but also the environment'? needs better phrasing

lines 497-504: You may want to discuss how various data-driven dietary inference methods (like those mentioned above) might be combined with your results, to provide a more complete/complementary view of dietary intake.

Reviewer #2

(Remarks to the Author)

This is a fascinating and timely paper by Fackelmann et al. I expect it to be highly cited by researchers working at the intersection of the microbiome and nutrition. In particular, its integration of so many samples from across multiple cohorts provide the level of epidemiological rigor needed to help us understand and believe the precise links structuring relationships between

the microbiome and nutrition.

Moderate to Major critiques:

* The magnitude of the AUCs distinguishing the three diets (Fig. 2) are very striking. I don't think these AUCs are outside the realm of possibility. Also, the distinguishing taxa (Figs 3 and 4) seem reasonable. Nevertheless, I'd want to be extremely sure of these AUCs before making such a high-profile claim. I'm particularly struck that the AUCs are so high for vegetarian vs. vegan. It does sound plausible that dairy-associated bacteria drive this classification ability. Still, it implies that a very high fraction of the sampled vegetarians have recently eaten dairy-based foods; could this really be true?

-- Part of double-checking these AUC values could include negative controls. It wasn't apparent to me from the Methods that these have been carried out. What metadata do the authors have access to that they would not expect there to be a microbiome association with? I'd suggest performing analyses against some of these non-microbiome associated dietary or lifestyle factors to ensure there hasn't been an error or bug in training (e.g. data leakage).

-- Another possibility explaining these AUCs is that the microbiome may be predicting another factor like geography or cohort membership, that is in turn also correlated with diet. For example, from Fig 1A, the ratio of mixed diets to vegans is higher in P1 than in Tarallo et al; it's possible the ML is "learning" to distinguish P1 to Tarallo. The low-to-moderate imbalance I see in Fig 1A make it seem unlikely that this effect could explain all of the AUC signal; still, can classifiers of similar AUC strength be constructed if training solely within one of the cohorts?

-- Another related thought: prior well-known studies (Rothschild, Nature 2018) that have linked diet to the microbiome still have only reported effect sizes, I believe, that are <0.2. While I understand and appreciate that ML/classification AUC scores are different from statistical analyses like PERMANOVA that report effect sizes, I think it would still be helpful in this paper to at least mirror some of the diet x overall microbiome structure PERMANOVA analyses from a paper like Rothschild, Nature 2018. Readers could then understand -- are the stronger relationships being reported here between nutrition and the microbiome due to analytical techniques (i.e. ML vs PERMANOVA); or, due to differences in the quality, composition, or size of the cohorts being used in this study.

Moderate Critiques:

* L409 -- Strep thermophilus' lower prevalence in the vegan cohorts makes sense. Still, presuming they are coming from dairy products, Fig 4i suggests Strep thermo is being consumed fairly frequently across both the mixed and vegetarian groups. Is intake of fermented dairy products really that common in those groups? My understanding is that these types of probiotic microbes only persist for 1-2 weeks; I'd be surprised if most mixed diet eaters actually consume say yogurt with that frequency.

I'm particularly struck that it's so high for vegetarian vs. vegan. Indeed, it sounds plausible that dairy-associated bacteria drive. Still, it implies that a very high fraction of the vegetarians have recently eaten dairy-based foods; unsure if that's actually true.

Minor critiques:

* L18 and L483 -- I'd be more cautious about the associations between diet-associated microbes and health. For example, Bilophila may not be playing causal roles in CRC or cardio metabolic disease; rather, it may serve as a biomarker of red meat intake, which in turn is responsible for deleterious effects on health. I also did not notice epidemiological analyses that attempt to quantify the distinct effects of specific microbes and diet vis a vis health (e.g. treating diet as a confounding variable when associating Bilophila to a given disease). Thus, I'd be more circumspect about implications between say probiotic/therapeutic potential of microbes and health.

* L126 -- If the data are available, it could be interesting to report how much of differences in richness could be associated with differences in transit time or Bristol stool score?

* L178 -- Ref to Fig 1b,h likely meant Fig 3?

* L277 -- This may be a critique for a paper in preparation (Asnicar et al., In prep), but how much of a microbe's health rank is actually being mediated by its prediction of dietary pattern (the latter variable actually being the causal variable for health)? To get at this, it would be interesting to do a correlation between 'Health Rank' and how strongly a given microbe predicts say a Vegan vs. Mixed Diet.

* L313 -- I'd be curious (and fine to place in supplement) examples of associations that cannot be predicted from FFQ. With datasets of this size, are there associations between all SGPs and foods?

* Fig 4J - Color bar is hard to read. White, it seems, is a middle score? Or, does it mean missing? I suggest a color gradient that doesn't move between colors and have white in the middle in order to read more easily

* I find it interesting to see that Prevotella, which has previously been strongly linked to high fiber diets across a number of global studies, does not appear to be one of the distinguishing taxa between the mixed and more plant-rich diets. Presuming there is room in the discussion for it, I'd be curious if these findings call into question that association.

* Do the authors have any information on how long people have been on a given diet? Such information could explain why some of the participants don't classify into their reported groups.

Decision Letter:

22nd May 2024

Dear Professor Segata,

Thank you for your patience while your manuscript "Microbiome signatures of vegan, vegetarian and mixed diets and links to health outcomes: a multi-population study of 21,561 individuals" was under peer-review at Nature Microbiology. It has now been seen by 2 referees, whose expertise and comments you will find at the end of this email. Although they find your work of some potential interest, they have raised a number of concerns that will need to be addressed before we can consider publication of the work in Nature Microbiology.

In particular, the code will need to be available to reviewers. Reviewer #1 thinks that you should compare and contrast your results with other existing approaches to dietary assessment, and that you should try to identify functional genes. Reviewer #2 has several questions about the AUCs, wants you to do PERMANOVA analyses to compare with other studies, and advice you to be more cautious about the associations between diet-associated microbes and health.

Should further experimental data/analyses allow you to address these criticisms, we would be happy to look at a revised manuscript.

Please include a data availability statement as a separate section after Methods but before references, under the heading "Data Availability". This section should inform readers about the availability of the data used to support the conclusions of your study. This information includes accession codes to public repositories (data banks for protein, DNA or RNA sequences, microarray, proteomics data etc...), references to source data published alongside the paper, unique identifiers such as URLs to data repository entries, or data set DOIs, and any other statement about data availability. At a minimum, you should include the following statement: "The data that support the findings of this study are available from the corresponding author upon request", mentioning any restrictions on availability. If DOIs are provided, we also strongly encourage including these in the Reference list (authors, title, publisher (repository name), identifier, year). For more guidance on how to write this section please see: <http://www.nature.com/authors/policies/data/data-availability-statements-data-citations.pdf>

* If you have not done so already we suggest that you begin to revise your manuscript so that it conforms to our Article format instructions at <http://www.nature.com/nmicrobiol/info/final-submission>. Refer also to any guidelines provided in this letter.

When submitting the revised version of your manuscript, please pay close attention to our [href="https://www.nature.com/nature-portfolio/editorial-policies/image-integrity">Digital Image Integrity Guidelines](https://www.nature.com/nature-portfolio/editorial-policies/image-integrity) and to the following points below:

Link Redacted

Note: This url links to your confidential homepage and associated information about manuscripts you may have submitted or be reviewing for us. If you wish to forward this e-mail to co-authors, please delete this link to your homepage first.

Nature Microbiology is committed to improving transparency in authorship. As part of our efforts in this direction, we are now requesting that all authors identified as 'corresponding author' on published papers create and link their Open Researcher and Contributor Identifier (ORCID) with their account on the Manuscript Tracking System (MTS), prior to acceptance. This applies to primary research papers only. ORCID helps the scientific community achieve unambiguous attribution of all scholarly contributions. You can create and link your ORCID from the home page of the MTS by clicking on 'Modify my Springer Nature account'. For more information please visit www.springernature.com/orcid.

If you wish to submit a suitably revised manuscript we would hope to receive it within 6 months. If you cannot send it within this time, please let us know. We will be happy to consider your revision, even if a similar study has been accepted for publication at Nature Microbiology or published elsewhere (up to a maximum of 6 months).

Yours sincerely,

Reviewer Expertise:

Referee #1: Microbiome and bioinformatics.

Referee #2: Microbiome and nutrition and machine learning.

Reviewer Comments:

Reviewer #1 (Remarks to the Author):

Summary -

The authors present a large cross-sectional study (N = 21,561) of how gut metagenomic features (microbial taxonomic units) are associated with dietary intake patterns. General dietary patterns were known for all of the individuals (656 vegans, 1,088 vegetarians, and 19,817 mixed diets), and more specific food frequency data was available for a subset (~13,000). The composition of the microbiome could classify dietary type (AUC = 0.85) and several taxonomic units were associated with specific foods and with host phenotypes. Not only did the authors identify human gut taxa that appeared to respond to the inclusion of specific foods, but they also found food-associated taxa that appear to serve as biomarkers of eating specific foods (e.g., dairy-associated lactobacilli or soil-associated taxa found on vegetables). This is an interesting and useful resource, presented in a well-written manuscript. I commend the authors on a nice paper. My major concerns involve the lack of detail on analysis code, along with a few minor concerns, outlined below.

Major Comments -

The authors do not provide a link to their analysis code (e.g., a GitHub repository). I see that they are using biobakery tools, but they need to provide the actual commands/code that was run to generate their results. There's also reference to 'custom scripts', which are not provided.

The thrust of this paper is to look for bacterial taxonomic biomarkers of dietary intake, but there are a few existing approaches to dietary assessment directly from data that are not discussed by the authors. It would be useful to compare and contrast their results to these other methods. For example, prior work has proposed using metabolomic biomarkers in stool to assess intake (<https://www.ncbi.nlm.nih.gov/pmc/articles/PMC9021190/>). Amplicon sequencing of chloroplasts in human stool has also been proposed (<https://www.pnas.org/doi/10.1073/pnas.2304441120>), in addition to direct detection of whole-genome food-related DNA in metagenomes (<https://www.biorxiv.org/content/10.1101/2024.02.02.578701v1>).

The authors focus on taxonomic biomarkers of dietary intake, but they don't try to identify functional genes. Why not? You get both sets of info from your metagenomes. It may be that functional gene profiles are more informative than taxonomic profiles.

Minor Comments -

line 18: change 'positively correlate with negative host' to 'negatively correlated with host'

line 28: 'risk' -> 'risk factor'

line 30-31: 'but also the environment'? needs better phrasing

lines 497-504: You may want to discuss how various data-driven dietary inference methods (like those mentioned above) might be combined with your results, to provide a more complete/complementary view of dietary intake.

Reviewer #2 (Remarks to the Author):

This is a fascinating and timely paper by Fackelmann et al. I expect it to be highly cited by researchers working at the intersection of the microbiome and nutrition. In particular, its integration of so many samples from across multiple cohorts provide the level of epidemiological rigor needed to help us understand and believe the precise links structuring relationships between the microbiome and nutrition.

Moderate to Major critiques:

* The magnitude of the AUCs distinguishing the three diets (Fig. 2) are very striking. I don't think these AUCs are outside the realm of possibility. Also, the distinguishing taxa (Figs 3 and 4) seem reasonable. Nevertheless, I'd want to be extremely sure of these AUCs before making such a high-profile claim. I'm particularly struck that the AUCs are so high for vegetarian vs. vegan. It does sound plausible that dairy-associated bacteria drive this classification ability. Still, it implies that a very high fraction of the sampled vegetarians have recently eaten dairy-based foods; could this really be true?

-- Part of double-checking these AUC values could include negative controls. It wasn't apparent to me from the Methods that these have been carried out. What metadata do the authors have access to that they would not expect there to be a microbiome association with? I'd suggest performing analyses against some of these non-microbiome associated dietary or lifestyle factors to ensure there hasn't been an error or bug in training (e.g. data leakage).

-- Another possibility explaining these AUCs is that the microbiome may be predicting another factor like geography or cohort membership, that is in turn also correlated with diet. For example, from Fig 1A, the ratio of mixed diets to vegans is higher in P1 than in Tarallo et al; it's possible the ML is "learning" to distinguish P1 to Tarallo. The low-to-moderate imbalance I see in Fig 1A make it seem unlikely that this effect could explain all of the AUC signal; still, can classifiers of similar AUC strength be constructed if training solely within one of the cohorts?

-- Another related thought: prior well-known studies (Rothschild, Nature 2018) that have linked diet to the microbiome still have only reported effect sizes, I believe, that are <0.2 . While I understand and appreciate that ML/classification AUC scores are different from statistical analyses like PERMANOVA that report effect sizes, I think it would still be helpful in this paper to at least mirror some of the diet x overall microbiome structure PERMANOVA analyses from a paper like Rothschild, Nature 2018. Readers could then understand -- are the stronger relationships being reported here between nutrition and the microbiome due to analytical techniques (i.e. ML vs PERMANOVA); or, due to differences in the quality, composition, or size of the cohorts being used in this study.

Moderate Critiques:

* L409 -- *Strep thermophilus*' lower prevalence in the vegan cohorts makes sense. Still, presuming they are coming from dairy products, Fig 4i suggests *Strep thermo* is being consumed fairly frequently across both the mixed and vegetarian groups. Is intake of fermented dairy products really that common in those groups? My understanding is that these types of probiotic microbes only persist for 1-2 weeks; I'd be surprised if most mixed diet eaters actually consume say yogurt with that frequency.

I'm particularly struck that it's so high for vegetarian vs. vegan. Indeed, it sounds plausible that dairy-associated bacteria drive. Still, it implies that a very high fraction of the vegetarians have recently eaten dairy-based foods; unsure if that's actually true.

Minor critiques:

* L18 and L483 -- I'd be more cautious about the associations between diet-associated microbes and health. For example, *Bifidobacteria* may not be playing causal roles in CRC or cardio metabolic disease; rather, it may serve as a biomarker of red meat intake, which in turn is responsible for deleterious effects on health. I also did not notice epidemiological analyses that attempt to quantify the distinct effects of specific microbes and diet vis a vis health (e.g. treating diet as a confounding variable when associating *Bifidobacteria* to a given disease). Thus, I'd be more circumspect about implications between say probiotic/therapeutic potential of microbes and health.

* L126 -- If the data are available, it could be interesting to report how much of differences in richness could be associated with differences in transit time or Bristol stool score?

* L178 -- Ref to Fig 1b,h likely meant Fig 3?

* L277 -- This may be a critique for a paper in preparation (Asnicar et al., In prep), but how much of a microbe's health rank is actually being mediated by its prediction of dietary pattern (the latter variable actually being the causal variable for health)? To get at this, it would be interesting to do a correlation between 'Health Rank' and how strongly a given microbe predicts say a Vegan vs. Mixed Diet.

* L313 -- I'd be curious (and fine to place in supplement) examples of associations that cannot be predicted from FFQ. With datasets of this size, are there associations between all SGPs and foods?

* Fig 4J - Color bar is hard to read. White, it seems, is a middle score? Or, does it mean missing? I suggest a color gradient that doesn't move between colors and have white in the middle in order to read more easily

* I find it interesting to see that *Prevotella*, which has previously been strongly linked to high fiber diets across a number of global studies, does not appear to be one of the distinguishing taxa between the mixed and more plant-rich diets. Presuming there is room in the discussion for it, I'd be curious if these findings call into question that association.

* Do the authors have any information on how long people have been on a given diet? Such information could explain why some of the participants don't classify into their reported groups.

Version 1:

Reviewer comments:

Reviewer #1

(Remarks to the Author)

I thank the authors for their thorough response to all reviewer concerns. The code availability section is now acceptable. The functional analysis adds a lot, and I appreciate their efforts. I commend them on an excellent manuscript. I have no further concerns, just a minor comment/clarification below.

I was surprised the authors ran the MEDI analysis, which is really great and truly above and beyond what I was expecting (especially given that it's still only a preprint, as they point out)! I'm sorry if I was unclear in my original review. I saw your current work as another attempt to build a data-driven method for classifying diet from microbiome compositional data for samples that lack FFQs or dietary questionnaire data. I have no doubt in the accuracy of your FFQs, and I think questionnaires are still the gold standard for dietary assessment. Data-driven methods are noisy, but they can provide additional information when we lack assessment data (and possibly, they might help correct certain biases inherent to questionnaires). When I made my comment, I was only asking the authors to discuss these alternative data-driven approaches and how these methods might be complementary, moving forward.

At this point, I might as well out myself as an author of the MEDI paper (Sean Gibbons). It's great to see that animal DNA is more prevalent in the omnivores. As for the other results, they make total sense given our experience. MEDI profiles are pretty sparse (I think of them like single-cell sequencing data). In your results, you may want to discuss power, in the context of not detecting animal DNA (or any other food item) in samples where you expect it. In our preprint, we find that you need at least 10 reads per million to get reliable detection of a given food item, and <0.1% of stool reads are diet-derived. Often, we find dietary profiles for metagenomes with fewer than ~30 million reads are too noisy to work with on a sample-by-sample basis (i.e., too much sparsity to interpret data coming from a single sample). It looks like your metagenomes were sequenced to depths of 10-20 million reads, which is below this depth level (we also see higher sparsity at these depths). In lower depth contexts, we see better results when pooling samples within diet groupings and looking at differences across groups (similar to what you did with animal DNA across the diets, which seems to work ok). We've also made some tweaks to the method during the review process that improves animal DNA detection (the final version of the method will have some improvements from what's presented in the preprint), but I don't expect you to rerun anything at this point (what you've already done is beyond what I expected). Anyway, thanks for trying out MEDI. Your results are consistent with what we're seeing. And I'm excited to explore if combining your microbiome signatures with our MEDI estimates might further improve data-driven dietary assessment/prediction (when lacking questionnaire data, which can be expensive to obtain and time consuming for study participants).

Reviewer #2

(Remarks to the Author)

Overall, I am satisfied with the authors' responses to my first round of critiques. Kudos to them for a thorough response.

I do have some comments though in regards to the authors' responses to Reviewer #1. I'll note that I typically do not engage in responses to other reviewers to minimize workload for the authors; however, in this case, I have some concerns with the authors' use/interpretation of MEDI, which could have an inadvertent and disproportionate effect on adoption of the tool since it is so young (in fact, I believe it is still a pre-print).

In short, I have concerns about the conclusion: "this tool nonetheless performed significantly worse than the FFQs in predicting participants' diet patterns (Chi-squared test, $p < 0.001$), with MEDI misclassifying 77% of participants."

My understanding is that Reviewer #1 wasn't asking for MEDI to be compared against the FFQ in terms of predicting diet. I did not see them question the veracity of FFQ. Their question was instead how MEDI compared against microbiome-based markers in terms of predicting diet. ("The thrust of this paper is to look for bacterial taxonomic biomarkers of dietary intake, but there are a few existing approaches to dietary assessment directly from data that are not discussed by the authors. It would be useful to compare and contrast their results to these other methods. For example, prior work has proposed using ...direct detection of whole-genome food-related DNA in metagenomes (<https://www.biorxiv.org/content/10.1101/2024.02.02.578701v1>)").

Furthermore, it seems unfair to consider simply presence/absence of total animal DNA as a decision rule for diet when the microbiome-based markers are using far more powerful ML techniques (e.g. RFs) to predict diet. If the authors choose to include the MEDI analyses in their paper, it seems fairer to ask how the same ML-based techniques used to predict diet perform when applied to the MEDI data.

Furthermore, I was struck by what the authors also noticed, which was that there was a much lower abundance of animal DNA in the vegetarians and vegans. I find it quite promising that MEDI could at the very least help confirm whether someone is indeed vegetarian or vegan. (I do see though that it doesn't look as promising for identifying omnivores from a single sample.)

In summary, my concern is that the author's new MEDI results could be mis-interpreted as a strongly negative result, when in reality, what was carried out was not designed to be a full and thorough validation of MEDI's potential and accuracy. I think it would be regrettable to inadvertently quash interest in this tool before it has had a chance to be fully explored and tested by the community.

Decision Letter:

Our ref: NMICROBIOL-24030957A

27th September 2024

Dear Dr. Segata,

Thank you for submitting your revised manuscript "Microbiome signatures of vegan, vegetarian and omnivore diets and links to health outcomes: a multi-population study of 21,561 individuals" (NMICROBIOL-24030957A). It has now been seen by the original referees and their comments are below. The reviewers find that the paper has improved in revision, and therefore we'll be happy in principle to publish it in Nature Microbiology, pending minor revisions to satisfy the referees' final requests and to comply with our editorial and formatting guidelines.

We are now performing detailed checks on your paper and will send you a checklist detailing our editorial and formatting requirements in a few weeks. Please do not upload the final materials and make any revisions until you receive this additional information from us.

Thank you again for your interest in Nature Microbiology Please do not hesitate to contact me if you have any questions.

Sincerely,

Reviewer #1 (Remarks to the Author):

I thank the authors for their thorough response to all reviewer concerns. The code availability section is now acceptable. The functional analysis adds a lot, and I appreciate their efforts. I commend them on an excellent manuscript. I have no further concerns, just a minor comment/clarification below.

I was surprised the authors ran the MEDI analysis, which is really great and truly above and beyond what I was expecting (especially given that its still only a preprint, as they point out)! I'm sorry if I was unclear in my original review. I saw your current work as another attempt to build a data-driven method for classifying diet from microbiome compositional data for samples that lack FFQs or dietary questionnaire data. I have no doubt in the accuracy of your FFQs, and I think questionnaires are still the gold standard for dietary assessment. Data-driven methods are noisy, but they can provide additional information when we lack assessment data (and possibly, they might help correct certain biases inherent to questionnaires). When I made my comment, I was only asking the authors to discuss these alternative data-driven approaches and how these methods might be complementary, moving forward.

At this point, I might as well out myself as an author of the MEDI paper (Sean Gibbons). It's great to see that animal DNA is more prevalent in the omnivores. As for the other results, they make total sense given our experience. MEDI profiles are pretty sparse (I think of them like single-cell sequencing data). In your results, you may want to discuss power, in the context of not detecting animal DNA (or any other food item) in samples where you expect it. In our preprint, we find that you need at least 10 reads per million to get reliable detection of a given food item, and <0.1% of stool reads are diet-derived. Often, we find dietary profiles for metagenomes with fewer than ~30 million reads are too noisy to work with on a sample-by-sample basis (i.e., too much sparsity to interpret data coming from a single sample). It looks like your metagenomes were sequenced to depths of 10-20 million reads, which is below this depth level (we also see higher sparsity at these depths). In lower depth contexts, we see better results when pooling samples within diet groupings and looking at differences across groups (similar to what you did with animal DNA across the diets, which seems to work ok). We've also made some tweaks to the method during the review process that improves animal DNA detection (the final version of the method will have some improvements from what's presented in the preprint), but I don't expect you to rerun anything at this point (what you've already done is beyond what I expected). Anyway, thanks for trying out MEDI. Your results are consistent with what we're seeing. And I'm excited to explore if combining your microbiome signatures with our MEDI estimates might further improve data-driven dietary assessment/prediction (when lacking questionnaire data, which can be expensive to obtain and time consuming for study participants).

Reviewer #2 (Remarks to the Author):

Overall, I am satisfied with the authors' responses to my first round of critiques. Kudos to them for a thorough response.

I do have some comments though in regards to the authors' responses to Reviewer #1. I'll note that I typically do not engage in responses to other reviewers to minimize workload for the authors; however, in this case, I have some concerns with the authors' use/interpretation of MEDI, which could have an inadvertent and disproportionate effect on adoption of the tool since it is so young (in fact, I believe it is still a pre-print).

In short, I have concerns about the conclusion: "this tool nonetheless performed significantly worse than the FFQs in predicting participants' diet patterns (Chi-squared test, $p < 0.001$), with MEDI misclassifying 77% of participants."

My understanding is that Reviewer #1 wasn't asking for MEDI to be compared against the FFQ in terms of predicting diet. I did not see them question the veracity of FFQ. Their question was instead how MEDI compared against microbiome-based markers in terms of predicting diet. ("The thrust of this paper is to look for bacterial taxonomic biomarkers of dietary intake, but there are a few existing approaches to dietary assessment directly from data that are not discussed by the authors. It would be useful to compare and contrast their results to these other methods. For example, prior work has proposed using ...direct detection of whole-genome food-related DNA in metagenomes (<https://www.biorxiv.org/content/10.1101/2024.02.02.578701v1>)").

Furthermore, it seems unfair to consider simply presence/absence of total animal DNA as a decision rule for diet when the microbiome-based markers are using far more powerful ML techniques (e.g. RFs) to predict diet. If the authors choose to include the MEDI analyses in their paper, it seems fairer to ask how the same ML-based techniques used to predict diet perform when applied to the MEDI data.

Furthermore, I was struck by what the authors also noticed, which was that there was a much lower abundance of animal DNA in the vegetarians and vegans. I find it quite promising that MEDI could at the very least help confirm whether someone is indeed vegetarian or vegan. (I do see though that it doesn't look as promising for identifying omnivores from a single sample.)

In summary, my concern is that the author's new MEDI results could be mis-interpreted as a strongly negative result, when in reality, what was carried out was not designed to be a full and thorough validation of MEDI's potential and accuracy. I think it would be regrettable to inadvertently quash interest in this tool before it has had a chance to be fully explored and tested by the community.

Version 2:

Decision Letter:

25th October 2024

Dear Professor Segata,

I am pleased to accept your Article "Gut microbiome signatures of vegan, vegetarian and omnivore diets and associated health outcomes across 21,561 individuals" for publication in Nature Microbiology. Thank you for having chosen to submit your work to us and many congratulations.

Please note that *Nature Microbiology* is a Transformative Journal (TJ). Authors may publish their research with us through the traditional subscription access route or make their paper immediately open access through payment of an article-processing charge (APC). Authors will not be required to make a final decision about access to their article until it has been accepted. Find out more about Transformative Journals

Authors may need to take specific actions to achieve compliance with funder and institutional open access mandates. If your research is supported by a funder that requires immediate open access (e.g. according to Plan S principles) then you should select the gold OA route, and we will direct you to the compliant route where possible. For authors selecting the subscription publication

route, the journal's standard licensing terms will need to be accepted, including [self-archiving policies](https://www.nature.com/nature-portfolio/editorial-policies/self-archiving-and-license-to-publish). Those licensing terms will supersede any other terms that the author or any third party may assert apply to any version of the manuscript.

With kind regards,

P.S. Click on the following link if you would like to recommend Nature Microbiology to your librarian <http://www.nature.com/subscriptions/recommend.html#forms>

** Visit the Springer Nature Editorial and Publishing website at [www.springernature.com/editorial-and-publishing-jobs](http://editorial-jobs.springernature.com?utm_source=ejP_NMicro_email&utm_medium=ejP_NMicro_email&utm_campaign=ejp_NMicro) for more information about our career opportunities. If you have any questions please click [here](mailto:editorial.publishing.jobs@springernature.com).**

use, sharing, adaptation, distribution and reproduction in any medium or format, as long as you give appropriate credit to the original author(s) and the source, provide a link to the Creative Commons license, and indicate if changes were made. In cases where reviewers are anonymous, credit should be given to 'Anonymous Referee' and the source. The images or other third party material in this Peer Review File are included in the article's Creative Commons license, unless indicated otherwise in a credit line to the material. If material is not included in the article's Creative Commons license and your intended use is not permitted by statutory regulation or exceeds the permitted use, you will need to obtain permission directly from the copyright holder. To view a copy of this license, visit <https://creativecommons.org/licenses/by/4.0/>

Nature Microbiology manuscript NMICROBIOL-24030957

“Microbiome signatures of vegan, vegetarian and omnivore diets and links to health outcomes: a multi-population study of 21,561 individuals”

Response to Editorial and Reviewers' comments

In this report letter we report the text from the editor and the reviewers with black text and our response in green. We are also submitted the paper with “track changes”.

Comments from the Editor:

Dear Professor Segata,

Thank you for your patience while your manuscript "Microbiome signatures of vegan, vegetarian and mixed diets and links to health outcomes: a multi-population study of 21,561 individuals" was under peer-review at Nature Microbiology. It has now been seen by 2 referees, whose expertise and comments you will find at the end of this email. Although they find your work of some potential interest, they have raised a number of concerns that will need to be addressed before we can consider publication of the work in Nature Microbiology.

In particular, the code will need to be available to reviewers. Reviewer #1 thinks that you should compare and contrast your results with other existing approaches to dietary assessment, and that you should try to identify functional genes. Reviewer #2 has several questions about the AUCs, wants you to do PERMANOVA analyses to compare with other studies, and advice you to be more cautious about the associations between diet-associated microbes and health.

Should further experimental data/analyses allow you to address these criticisms, we would be happy to look at a revised manuscript.

Answer: We thank the editor and the reviewers for their time and for providing such detailed feedback. We are glad to note the overall appreciation of our work. In this revision, we have addressed all major concerns raised by the reviewers. Here, we briefly summarize the major points we addressed and that were raised by reviewers:

- We have added the code used in this analysis in the new **Supplementary File 1** and included a corresponding statement in the Code Availability section of this manuscript, where we also link to any code used and available on GitHub.
- We made considerable effort to apply a novel, unpublished tool suggested by reviewers (MEDI) to estimate dietary intake from food DNA in stool samples and compared results

herefrom to our results based on detailed food frequency questionnaires (FFQs). We find that while MEDI can provide valuable information, it is not accurate in distinguishing between vegetarians and omnivores, since both their guts may contain non-human DNA of animal origin. Moreover, even with a simplified classification of vegan vs non-vegans, MEDI still misclassified 77% of participants. Thus, at present and for our study in particular, we conclude MEDI can provide some valuable information for some foods of interest, but cannot substitute or reliably support FFQ data.

- We conducted a set of microbial metabolic pathway-level analyses aimed at identifying the functional modules enriched in the microbiome communities of the three diet patterns. These further supported our taxonomically-based conclusions by showing that functions enriched in vegans and vegetarians are commonly present in microbes inhabiting plant-rich niches, whereas functions enriched in omnivore metagenomes are involved in the breakdown of animal-derived foods such as red and white meat as well as dairy.
- We clarified and added further details to describe the various machine learning methods we implemented to ensure robust AUC values and added some experiments that further show that results were not affected by data leakage issues.
- We show that 96% of omnivores and 90% vegetarians do indeed consume dairy at least once a week, which supports the notion that our results concerning the links between dairy-associated bacteria and the gut microbiome are not erroneous.

Point-by-point replies to all comments are presented in the rest of this rebuttal letter. By addressing all these concerns, we hope the editor and reviewers find this work to be suitable for publication in *Nature Microbiology*.

General comments:

Please note that we made minor changes like changing the citation style to match *Nature Microbiology's* guidelines and making the font size in Figure 4a larger. We were made aware that the cohorts P3 22UKA and P3 22USA should be called P3 UK22A and P3 US22A respectively, which we also changed throughout. After some discussion, we also decided that the term “omnivore” is more appropriate and fits into the current literature better than “mixed diet”, so we’ve changed this term throughout the manuscript, figures, and supplementary files.

Comments from Reviewer #1 (Remarks to the Author):

Summary -

The authors present a large cross-sectional study (N = 21,561) of how gut metagenomic features (microbial taxonomic units) are associated with dietary intake patterns. General dietary patterns were known for all of the individuals (656 vegans, 1,088 vegetarians, and 19,817 mixed diets), and more specific food frequency data was available for a subset (~13,000). The composition of the microbiome could classify dietary type (AUC = 0.85) and several taxonomic units were associated with specific foods and with host phenotypes. Not only did the authors identify human gut taxa that appeared to respond to the inclusion of specific foods, but they also found food-associated taxa that appear to serve as biomarkers of eating specific foods (e.g., dairy-associated lactobacilli or soil-associated taxa found on vegetables). This is an interesting and useful resource, presented in a well-written manuscript. I commend the authors on a nice paper. My major concerns involve the lack of detail on analysis code, along with a few minor concerns, outlined below.

Answer: We thank the reviewer for their positive feedback and constructive comments, which we hope to have satisfyingly addressed in this revised manuscript. Each point is addressed in detail below.

Major Comments -

The authors do not provide a link to their analysis code (e.g., a GitHub repository). I see that they are using biobakery tools, but they need to provide the actual commands/code that was run to generate their results. There's also reference to 'custom scripts', which are not provided.

Answer: We thank the reviewer for picking up on our mistake: We have provided the code in **Supplementary File 1** and GitHub links in the Data Availability Statement. We reference the new **Supplementary File 1** whenever applicable, but highlight here the example the reviewer referred to in the Methods section as follows:

“The pooled estimate of effect sizes from linear models implements a random-effects meta-analysis with DerSimonian-Laird heterogeneity (DerSimonian and Laird 1986) (**Supplementary File 1; Data Availability Statement**).”

as well as in the Data Availability section as follows:

“The code for the analyses conducted here is provided in **Supplementary File 1**. The pooled estimate of effect sizes from linear models was computed based on the pipeline https://github.com/waldronlab/curatedMetagenomicDataAnalyses/blob/main/python_tools/metaanalyze.py. An importable meta-analysis python library is also

freely available at
https://github.com/SegataLab/inverse_var_weight/blob/main/meta_analyses.py.”

The thrust of this paper is to look for bacterial taxonomic biomarkers of dietary intake, but there are a few existing approaches to dietary assessment directly from data that are not discussed by the authors. It would be useful to compare and contrast their results to these other methods. For example, prior work has proposed using metabolomic biomarkers in stool to assess intake (<https://www.ncbi.nlm.nih.gov/pmc/articles/PMC9021190/>). Amplicon sequencing of chloroplasts in human stool has also been proposed (<https://www.pnas.org/doi/10.1073/pnas.2304441120>), in addition to direct detection of whole-genome food-related DNA in metagenomes (<https://www.biorxiv.org/content/10.1101/2024.02.02.578701v1>).

Answer: We thank the reviewer for sharing these resources and ideas with us. We agree with the reviewer in that several of our results highlight bacterial taxa that have been described in the literature as linked with the consumption of certain foods and which are shared between gut and food microbiomes. Although we have high confidence in our FFQs that have been extensively validated over time and in publications (Asnicar, Berry, et al. 2021; Berry et al. 2020; Mazidi et al. 2021; Bermingham et al. 2023), we nonetheless, made a considerable effort to run the MEDI tool (Diener and Gibbons 2024), since we had no metabolomics or amplicon data for the other methods. Doing so, we first noticed that there was a lower prevalence of animal DNA among vegans vs non-vegans (new **Supplementary Figure 7**; see below), which encouraged us to assess how a MEDI-based classification of diet patterns would perform versus an FFQ-based classification, using what participants self-reported as the ground truth (only participants from P1, P3 22UKA, and P3 22USA were considered, since these are the only cohorts with FFQs and all three diet patterns). To do so, we classified any sample that MEDI found animal DNA in as a non-vegan sample and any sample that no animal DNA was found in as a vegan sample. Similarly, we classified any sample whose FFQ reported the consumption of any animal product as a non-vegan sample and vice versa. We found MEDI to perform significantly worse than our FFQs in predicting participants' diet patterns (Chi-squared test, $p < 0.001$) with MEDI misclassifying 77% of participants.

Similarly, MEDI can only capture food that was recently ingested and not long-term diet patterns. In addition, only food types whose DNA is able to survive cooking, processing, and the digestive tract will be detectable. These are limitations that the authors of MEDI themselves discuss and acknowledge in their pre-print. All of these limitations made themselves visible in that animal DNA was found in only 21% of non-vegans, which contradicts participants' self-reported diet patterns and their FFQs, and suggests that the majority of animal DNA degrades during digestion, thus not being detectable in feces.

Moreover, MEDI is not able to reliably distinguish between vegetarians and omnivores, since it is possible for both their guts to contain animal DNA. While this issue can be mitigated by thresholding the MEDI read mapping with higher stringency, it would have an opposite effect on the omnivore analysis in which we already detect a very low amount of animal DNA (**Supplementary Figure 7**). Thus, MEDI cannot be used to classify individuals into omnivores, vegetarians, or vegans using single cross-sectional metagenomic samples.

We acknowledge that this tool can certainly be useful in studies lacking FFQ data, however, when FFQ data are available, we find this method to underperform and hope the reviewer agrees. We highlighted these results briefly in the Results and Methods sections and included this figure and its table in the Supplement (new **Supplementary Figure 7**; new **Supplementary Table 24**). The Results now reads as follows:

“This information and FFQs proved much more accurate than DNA-based detection of food components in the stool microbiome (Diener and Gibbons 2024) (Methods) and enabled further characterization of the food-microbiome relationship at the single food-type resolution within each diet pattern.”

The Methods now reads as follows:

“To further support the FFQs, we tested the Metagenomic Estimation of Dietary Intake (MEDI) tool (Diener and Gibbons 2024), which uses food DNA in gut metagenomes to estimate and quantify food consumption (<https://github.com/Gibbons-Lab/medi>). We assessed how a MEDI-based classification of diet patterns would perform versus an FFQ-based classification, using what participants self-reported as the ground truth (only participants from P1, P3 22UKA, and P3 22USA were considered, since these are the only cohorts with FFQs and all three diet patterns). To do so, we classified any sample in which MEDI found animal DNA as a non-vegan sample and any sample in which no animal DNA was found as a vegan sample. Similarly, we classified any sample whose FFQ reported the consumption of any animal product as a non-vegan sample and vice versa. While there was a lower prevalence of animal DNA among vegans vs non-vegans (**Supplementary Figure 7**; **Supplementary Table 24**) using the MEDI classification, this tool nonetheless performed significantly worse than the FFQs in predicting participants’ diet patterns (Chi-squared test, $p < 0.001$), with MEDI misclassifying 77% of participants. Due to this lesser accuracy and reliability compared to FFQs, we opted to base any analyses using food consumption data on the FFQs that have been extensively validated over time and in publications (Asnicar, Berry, et al. 2021; Berry et al. 2020; Mazidi et al. 2021; Bermingham et al. 2023) instead of MEDI estimates.”

Supplementary Figure 7. Prevalence (in %) of animal DNA as estimated by MEDI using the gut metagenomes of omnivores (bars in pink), vegetarians (bars in purple), and vegans (bars in green) in the P1, P3 UK22A and P3 US22A cohorts (**Supplementary Table 24**).

The authors focus on taxonomic biomarkers of dietary intake, but they don't try to identify functional genes. Why not? You get both sets of info from your metagenomes. It may be that functional gene profiles are more informative than taxonomic profiles.

Answer: We thank the reviewer for their insightful suggestion, which we have followed. We generated microbial functional profiles using HUMAnN version 3.6, which we meta-analyzed in keeping with the current methodology of the manuscript. We decided to focus on the profiling of pathway-level functional characteristics as including information on single genes would have required substantially more work and space in the manuscript.

The results are detailed in the section '**Gut microbial functional potential associated with diet patterns supports plant- and meat-specific niches**' as follows:

Since our results pointed toward gut microbial configurations seemingly adapted to the ingestion of major food groups, we explored this hypothesis by looking at the diet patterns' gut microbial functional potential as determined by HUMAnN version 3.6. Specifically, we focused on the whole-pathway level profiles that were meta-analyzed similar to the taxonomic profiles (**Methods**). This revealed an array of plant-associated microbial functions that were enriched in vegetarian and vegan diets compared to an omnivore diet. These included the conversion of simple carbohydrates (e.g., D-galactose degradation-pathway 6317) and of bioactive compounds (**Supplementary Figures 4-6**; **Supplementary Tables 21-23**). Among the latter we identified the *myo*-, *chiro*- and *scillo*-inositol degradation pathway 7237, whose molecules (i.e., the bioactive forms of inositol/vitamin B₇) represent the most abundant and accessible carbon and energy sources in plant-associated environments (Weber and Fuchs 2022). This

pathway is particularly widespread across soil and rhizosphere bacteria and may therefore provide a competitive advantage for growth and substrate utilization to microbes in plant-associated niches (Weber and Fuchs 2022), such as those of a vegan or vegetarian gut microbiome. Also enriched in vegan and vegetarian gut microbes were pathways for the biosynthesis of chorismate (ARO and 6163), an intermediate in the production of various essential metabolites (e.g., some aromatic amino acids, vitamins E and K, and ubiquinone (Hoch and Nester 1973)). These enzymatic routes are of note because they are shared among prokaryotes, including plant endosymbiotic cyanobacteria (Hoch and Nester 1973), as well as several eukaryotes, including ascomycete fungi, plants and algae (Dosselaere and Vanderleyden 2001; Richards et al. 2006). This underscores yet again the enrichment of functions associated with plant ecological niches in vegan and vegetarian gut metagenomes.

In contrast, pathways overrepresented in omnivore gut microbiomes were involved in the breakdown of animal-derived foods and amino acid metabolism (**Supplementary Figures 4-6; Supplementary Tables 21-23**). These included the superpathway of L-threonine (THRESYN), and of L-serine and glycine biosynthesis (SER-GLYSYN), whose substrates are commonly found in red and white meat, and in dairy products (Chassagnole et al. 2001; Handzlik and Metallo 2023). Moreover, we found that omnivore microbiomes displayed the enzymatic machinery necessary for the salvage of essential cofactors that are abundant in foods of animal origin, such as organ meats, shellfish, and eggs (Watanabe et al. 2013; Carmel 2005). These cofactors included adenosylcobalamin (vitamin B₁₂) and folate (vitamin B₉) from dietary precursors (e.g., cobinamide in the COBALSYN pathway and 10-formyl-tetrahydrofolate in the 1CMET2 pathway, respectively (de Crécy-Lagard et al. 2007; Nagy et al. 1995)). The former is of particular note, since its precursors are derived from animal sources absent in vegan diets, making vitamin B₁₂ supplementation necessary for vegans (Green et al. 2017). In summary, gut microbial functional potential reveals diet-specific niches related to the metabolism of animal- or plant-derived foods, supporting the role of diet and its inclusion or exclusion of major food groups in shaping the gut microbial landscape both taxonomically and functionally.

Moreover, results are discussed in the penultimate paragraph of the **Discussion** as follows:

Evidence for the selection of microbes according to one's overall dietary pattern was reflected not only taxonomically but also functionally. Microbial pathways that confer competitive advantages in plant-rich environments were enriched in vegans and vegetarians compared to omnivores. Conversely, pathways involved in the breakdown of animal-derived foods were enriched in omnivores. For instance, the vitamin B₁₂ precursor of the enriched COBALSYN pathway is derived from animal sources and thus absent in vegan diets, making vitamin B₁₂ supplementation necessary for vegans but not for omnivores with this

metagenomic pathway (Green et al. 2017). The mechanisms of diet-specific functional enrichment via species selection or via acquisition of functional modules is one of the aspects that will need further dedicated lines of research and would require functional analyses resolved at the level of single strains.

In addition, the methods are described under the sub-header '**Gut microbial functional potential**' as follows:

To generate gut microbial functional potential, we ran HUMAnN version 3.6 (Beghini et al. 2021) with default parameters (**Supplementary File 1**). We focused on the pathway abundance output and removed any unmapped or unintegrated pathways, as well as pathways with a prevalence of less than 0.05 across samples of at least one diet pattern and a coverage less than 0.2. This left us with 87 pathways in P1, 85 pathways in P3 UK22A, and 87 pathways in P3 US22A. We then measured the statistical association between the relative abundances of these pathways and each diet pattern pair, which we first computed on each of the three PREDICT cohorts separately and then meta-analyzed as described in detail above (**Supplementary File 1**).

Minor Comments -

line 18: change 'positively correlate with negative host' to 'negatively correlated with host'

Answer: We thank the reviewer for picking up on this mistake, which we have corrected as suggested.

line 28: 'risk' -> 'risk factor'

Answer: We thank the reviewer for picking up on this mistake, which we have corrected as suggested.

line 30-31: 'but also the environment'? needs better phrasing

Answer: We thank the reviewer for their suggestion and have rephrased this sentence as follows:

“Unhealthy diets not only increase the global burden on non-communicable diseases, but also carry a wide range of negative environmental impacts (Willett et al. 2019).”

lines 497-504: You may want to discuss how various data-driven dietary inference methods (like those mentioned above) might be combined with your results, to provide a more complete/complementary view of dietary intake.

Answer: We thank the reviewer for sharing these resources and ideas with us. We agree with the reviewer in that several of our results highlight bacterial taxa that have been described in the literature as linked with the consumption of certain foods and which are shared between gut and food microbiomes. Although we have high confidence in our FFQs that have been extensively validated over time and in publications (Asnicar, Berry, et al. 2021; Berry et al. 2020; Mazidi et

al. 2021; Bermingham et al. 2023), we nonetheless, made a considerable effort to run the MEDI tool (Diener and Gibbons 2024), since we had no metabolomics or amplicon data for the other methods. Doing so, we first noticed that there was a lower prevalence of animal DNA among vegans vs non-vegans (new **Supplementary Figure 7**; see below), which encouraged us to assess how a MEDI-based classification of diet patterns would perform versus an FFQ-based classification, using what participants self-reported as the ground truth (only participants from P1, P3 22UKA, and P3 22USA were considered, since these are the only cohorts with FFQs and all three diet patterns). To do so, we classified any sample that MEDI found animal DNA in as a non-vegan sample and any sample that no animal DNA was found in as a vegan sample. Similarly, we classified any sample whose FFQ reported the consumption of any animal product as a non-vegan sample and vice versa. We found MEDI to perform significantly worse than our FFQs in predicting participants' diet patterns (Chi-squared test, $p < 0.001$) with MEDI misclassifying 77% of participants.

Similarly, MEDI can only capture food that was recently ingested and not long-term diet patterns. In addition, only food types whose DNA is able to survive cooking, processing, and the digestive tract will be detectable. These are limitations that the authors of MEDI themselves discuss and acknowledge in their pre-print. All of these limitations made themselves visible in that animal DNA was found in only 21% of non-vegans, which contradicts participants' self-reported diet patterns and their FFQs, and suggests that the majority of animal DNA degrades during digestion, thus not being detectable in feces.

Moreover, MEDI is not able to reliably distinguish between vegetarians and omnivores, since it is possible for both their guts to contain animal DNA. While this issue can be mitigated by thresholding the MEDI read mapping with higher stringency, it would have an opposite effect on the omnivore analysis in which we already detect a very low amount of animal DNA (**Supplementary Figure 7**). Thus, MEDI cannot be used to classify individuals into omnivores, vegetarians, or vegans using single cross-sectional metagenomic samples.

We acknowledge that this tool can certainly be useful in studies lacking FFQ data, however, when FFQ data are available, we find this method to underperform and hope the reviewer agrees. We highlighted these results briefly in the Results and Methods sections and included this figure and its table in the Supplement (new **Supplementary Figure 7**, new **Supplementary Table 24**). The Results now reads as follows:

“This information and FFQs proved much more accurate than DNA-based detection of food components in the stool microbiome (Diener and Gibbons 2024) (Methods) and enabled further characterization of the food-microbiome relationship at the single food-type resolution within each diet pattern.”

The Methods now reads as follows:

“To further support the FFQs, we tested the Metagenomic Estimation of Dietary Intake (MEDI) tool (Diener and Gibbons 2024), which uses food DNA in gut metagenomes to estimate and quantify food consumption (<https://github.com/Gibbons-Lab/medi>). We assessed how a MEDI-based classification of diet patterns would perform versus an FFQ-based classification,

using what participants self-reported as the ground truth (only participants from P1, P3 22UKA, and P3 22USA were considered, since these are the only cohorts with FFQs and all three diet patterns). To do so, we classified any sample in which MEDI found animal DNA as a non-vegan sample and any sample in which no animal DNA was found as a vegan sample. Similarly, we classified any sample whose FFQ reported the consumption of any animal product as a non-vegan sample and vice versa. While there was a lower prevalence of animal DNA among vegans vs non-vegans (**Supplementary Figure 7, Supplementary Table 24**) using the MEDI classification, this tool nonetheless performed significantly worse than the FFQs in predicting participants' diet patterns (Chi-squared test, $p < 0.001$), with MEDI misclassifying 77% of participants. Due to this lesser accuracy and reliability compared to FFQs, we opted to base any analyses using food consumption data on the FFQs instead of MEDI estimates.”

Supplementary Figure 7. Prevalence (in %) of animal DNA as estimated by MEDI using the gut metagenomes of omnivores (bars in pink), vegetarians (bars in purple), and vegans (bars in green) in the P1, P3 UK22A and P3 US22A cohorts (**Supplementary Table 24**).

Comments from Reviewer #2 (Remarks to the Author):

This is a fascinating and timely paper by Fackelmann et al. I expect it to be highly cited by researchers working at the intersection of the microbiome and nutrition. In particular, its integration of so many samples from across multiple cohorts provide the level of epidemiological rigor needed to help us understand and believe the precise links structuring relationships between the microbiome and nutrition.

Answer: We thank the reviewer for their positive and encouraging feedback as well as their constructive comments, which we have addressed point-by-point below.

Moderate to Major critiques:

* The magnitude of the AUCs distinguishing the three diets (Fig. 2) are very striking. I don't think these AUCs are outside the realm of possibility. Also, the distinguishing taxa (Figs 3 and 4) seem reasonable. Nevertheless, I'd want to be extremely sure of these AUCs before making such a high-profile claim. I'm particularly struck that the AUCs are so high for vegetarian vs. vegan. It does sound plausible that dairy-associated bacteria drive this classification ability. Still, it implies that a very high fraction of the sampled vegetarians have recently eaten dairy-based foods; could this really be true?

Answer: We fully understand the reviewer and were also cautiously optimistic about these AUC values. Indeed, dairy-associated microbes such as *Streptococcus thermophilus* were among the most predictive in distinguishing between vegans and vegetarians (**Figure 4b**), so our machine learning results are strongly supported by our other results and methods as well. Moreover, we find the highest cumulative relative abundance of dairy SGBs in vegetarians (even higher than in omnivores, as can be seen in **Figure 4g**). We don't find this to be out of the realm of possibility since vegetarianism is distinguished from veganism almost purely on the basis of dairy consumption, since both groups do not eat meat but do eat fruits and vegetables. Very anecdotally, we find a common reason given by vegetarians for not being vegan is their love for and inability to give up cheese.

To support this, we further looked, in this revised manuscript, at responses from the FFQs for frequency of consumption of dairy products (milk, yogurt, cheese, butter, other dairy) in both UK and US PREDICT3 cohorts. We find that 96% of omnivores and 90% of vegetarians consume dairy at least once per week (see figure below, which we've added to the manuscript as **Supplementary Figure 3**). Both omnivores and vegetarians report very similar (though not identical) dairy consumption frequencies, with omnivores reporting more consumption at the higher frequencies (FFQ responses "2 or more times a day", "once a day", and "5-6 times a week"). Even if we consider only fermented dairy products (yogurt, cheeses; see figure below), we still find that the vast majority of omnivores (90%) and vegetarians (84%) consume fermented dairy products at least once a week. The biggest difference in consumption of fermented dairy is that there is less consumption on a daily basis in favor of more consumption during the week. Interestingly, we observed that vegetarians consume fermented dairy more frequently on a daily basis than omnivores, which we think fits nicely with what we know anecdotally about vegetarian dairy consumption. Regardless of whether we consider all dairy products or only fermented dairy, we conclude that both omnivores and vegetarians consume

dairy often enough to support our findings that dairy-associated bacteria drive our classification ability. We added these findings to the manuscript under the **Results** section “**Microbial species shared between the gut and food differ across diet patterns**” as follows:

“To lend more support to this hypothesis, we assessed omnivore and vegetarian frequency of dairy consumption (milk, yogurt, cheese, butter, other dairy) according to the FFQS in both P3 UK22A and P3 US22A cohorts. We found that 96% of omnivores and 90% of vegetarians consume dairy at least once per week (**Supplementary Figure 3**) and both diet patterns report very similar (though not identical) dairy consumption frequencies, with omnivores consuming dairy at the higher frequencies (FFQ responses “2 or more times a day”, “once a day”, and “5-6 times a week”). Even if we consider only fermented dairy products (yogurt and cheeses; **Supplementary Figure 3**), we still find that the vast majority of omnivores (90%) and vegetarians (84%) consume fermented dairy products at least once per week. Thus, we conclude that, while some microbes signature of diets that include dairy consumption could be selected to help digest dairy, others could be present in the gut microbiome as transient members derived from dairy foods themselves.”

Furthermore, it should be noted that, in both the UK and US, dairy consumption is high (OECD and Food and Agriculture Organization of the United Nations 2021), consistent with public health recommendations in both countries advocating daily consumption of dairy as part of a healthy balanced diet (Public Health England. 2016; U.S. Department of Agriculture and U.S. Department of Health and Human Services. 2020). Due to the absence of meat-based food groups in vegetarian diets, it is likely that dairy intake in this group is high to account for the lack of meat-based sources of protein. Based on the results we have presented in combination with population intakes outlined, we are confident that our sample likely consumes dairy on a regular basis, and that this is reflected in our findings.

Supplementary Figure 3 Frequency of dairy consumption across omnivores and vegetarians in P3 UK22A and P3 US22A according to FFQs. a Percentage (y-axis) of omnivores and vegetarians (x-axis) that consume dairy products (milk, yogurt, cheese, butter, or other dairy) between “two or more times per day” to “irregularly”. The consumption frequency categories were given by the FFQs. Percentages within the bar plots indicate the dairy consumption prevalence of that diet pattern in that consumption category. **b** Same as in a, but considering only fermented dairy products (yogurt and cheeses).

-- Part of double-checking these AUC values could include negative controls. It wasn't apparent to me from the Methods that these have been carried out. What metadata do the authors have access to that they would not expect there to be a microbiome association with? I'd suggest performing analyses against some of these non-microbiome associated dietary or lifestyle factors to ensure there hasn't been an error or bug in training (e.g. data leakage).

Answer: We thank the reviewer for their suggestion and thoroughness in assessing our manuscript. Data leakage is of course an important issue in machine learning and one we take very seriously. Before we mention the experiments that we ran to confirm the validity of our results, we would like to highlight that this method using our metAML tool has been validated and confirmed across a wide range of successful publications (Thomas et al. 2019; Beghini et al. 2021; Tarallo et al. 2019; Manghi et al. 2023; Valles-Colomer et al. 2024; Lee et al., n.d.; Ghensi et al. 2020) to be robust to such an extent that we consider it a validation tool for the robustness of other similar softwares. This is in part because the code that handles the training

and testing sets separation does not change due to an optimal implementation based on a high-dimensional boolean matrix, thus preventing data leakage by virtue of a binary (True/False) logic. We specifically discuss these aspects in a recent review paper (Asnicar et al. 2024).

Nonetheless, we took the reviewer's suggestion to heart and came up with what we hope is a satisfactory solution. Since we did not measure any metadata variables that we can expect *a-priori* to be entirely non-associated with the microbiome (since those would not be of interest and thus unnecessarily costly to measure in the first place), we instead opted to model an *ad-hoc* unrelated variable that we created via random shuffling of the diet pattern classes (omnivore, vegetarian, vegan) across all the individuals, which eliminates the natural link between microbiome and diet. This was done prior to the training/testing phases. With these shuffled classes, we then ran the same analysis in the same way as reported in the main **Figure 2**. In this experimental case, the presence of data leakage would be revealed by a prediction power significantly above AUC = 0.5. The following figure compares our original results (with the true diet pattern assignment; solid lines) with the new re-shuffling results (dotted lines). The average AUCs across the cohorts are reported in the color legend, with the first AUC corresponding to the original analysis and the second AUC corresponding to the shuffled analysis. From this and the graphs, it is clear that there was no data leakage. Based on these experiments and this tool's extensive publication history, we hope to have reassured the reviewer of the validity of our machine learning results. We mentioned this experiment in the Methods section, where we describe our machine learning approach as follows:

“To test for potential data leakage and overfitting, we randomly swapped the diet pattern labels in a cross-LODO experiment, which, as expected, did not result in AUCs above 0.51.”

-- Another possibility explaining these AUCs is that the microbiome may be predicting another factor like geography or cohort membership, that is in turn also correlated with diet. For example, from Fig 1A, the ratio of mixed diets to vegans is higher in P1 than in Tarallo et al; it's possible the ML is "learning" to distinguish P1 to Tarallo. The low-to-moderate imbalance I see in Fig 1A make it seem unlikely that this effect could explain all of the AUC signal; still, can classifiers of similar AUC strength be constructed if training solely within one of the cohorts?

Answer: In order to prevent exactly this, we had also conducted a LODO to train on all but one cohort and test on the left-out cohort, i.e., the cohort which the model has never seen before. In this manner, our ML models will never see the testing (“left-out”) cohort during training. Even doing so, our average AUC values are as follows (Supplementary Table 9): 0.81 omnivore-vegetarian; 0.89 omnivore-vegan; 0.81 vegetarian-vegan. This was described in the Methods as follows:

“We performed three types of validation: i) per-cohort, ten-times repeated, ten-fold cross-validation; ii) leave-one-dataset-out (LODO), which consist in a training which encompasses all cohorts but one and is tested on the left-out cohort (and iteratively done for all cohorts); iii) a hybrid approach named cross-LODO, which corresponds to a per-cohort (ten-times, ten-folds)

cross-validation, in which the rest of the cohorts are added to each training set as a support.”

However and to ensure clarity, we have now added this information to the Results section as well:

“The highest predictability was obtained when separating the most different diets, namely vegans from omnivores (mean cross-LODO AUC across all cohorts = 0.90), followed by vegetarians and vegans (mean cross-LODO AUC across all cohorts = 0.84), and finally vegetarians and omnivores (mean cross-LODO AUC across all cohorts = 0.82; **Figure 2; Supplementary Table 9**). Similar results were achieved when using the LODO approach (**Supplementary Table 9**) that does consider any training folds from the target cohort.”

-- Another related thought: prior well-known studies (Rothschild, Nature 2018) that have linked diet to the microbiome still have only reported effect sizes, I believe, that are <0.2. While I understand and appreciate that ML/classification AUC scores are different from statistical analyses like PERMANOVA that report effect sizes, I think it would still be helpful in this paper to at least mirror some of the diet x overall microbiome structure PERMANOVA analyses from a paper like Rothschild, Nature 2018. Readers could then understand -- are the stronger relationships being reported here between nutrition and the microbiome due to analytical techniques (i.e. ML vs PERMANOVA); or, due to differences in the quality, composition, or size of the cohorts being used in this study.

Answer: The reviewer raises a good point, we certainly want to ensure comparability between these newer results and previous ones. We had already included PERMANOVA results in the Results section as follows:

“Overall gut microbial composition (microbial diversity between individuals; beta diversity) as inferred by MetaPhlAn 4’s species-level genome bin (SGB) identification and quantification (Blanco-Míguez, Beghini, et al. 2023) (see **Methods**) also differed significantly according to diet pattern (permutational multivariate analysis of variance, PERMANOVA, on unweighted UniFrac distances, $R^2 = 0.002$ to 0.028 ; $p < 0.05$ for all five cohorts; **Figures 1d-h; Supplementary Table 8** also with additional distance metrics; **Methods**).”

This included the R^2 of 0.002 to 0.028 (depending on cohort) which corresponds to the effect of diet pattern on gut microbial beta diversity using unweighted UniFrac distances, but, as mentioned here, we also report the R^2 for weighted UniFrac, Bray-Curtis, and Aitchison distances in **Supplementary Table 8**. To compare with Rothschild, Nature 2018, they stated: “20% of the inter-person microbiome variability is associated with factors related to diet, drugs and anthropometric measurements.”, so presumably the 0.2 that the reviewer mentioned refers to this. We would like to point out, however, that this 20% is a combination of factors and not just the diet contribution. According to Extended Data Figure 4 in Rothschild, Nature 2018, “Diet” explains roughly 0.0125 of variance using what we presume are Bray-Curtis distances, though

this is not explicitly stated here. Nonetheless, this corresponds to our R² values for Bray-Curtis distances which range between 0.001 and 0.023. We have added this information to the manuscript and hope that the reviewer agrees that we have included sufficient statistical information in the manuscript to ensure that readers are able to compare results from other studies with ours. This section now reads as follows:

“Overall gut microbial composition (microbial diversity between individuals; beta diversity) as inferred by MetaPhlAn 4’s species-level genome bin (SGB) identification and quantification (Blanco-Míguez, Beghini, et al. 2023) (see **Methods**) also differed significantly according to diet pattern (permutational multivariate analysis of variance, PERMANOVA, on unweighted UniFrac distances, R² = 0.002 to 0.028; $p < 0.05$ for all five cohorts; **Figures 1d-h**; **Supplementary Table 8** also with additional distance metrics; **Methods**). The variation in beta diversity explained by diet pattern aligns with previous studies (compare our R² values for Bray-Curtis distances that range between 0.001 and 0.023 depending on cohort with 0.0125 reported by Rothschild and colleagues (2018) (Rothschild et al. 2018)).”

Moderate Critiques:

* L409 -- Strep thermophilus' lower prevalence in the vegan cohorts makes sense. Still, presuming they are coming from dairy products, Fig 4i suggests Strep thermo is being consumed fairly frequently across both the mixed and vegetarian groups. Is intake of fermented dairy products really that common in those groups? My understanding is that these types of probiotic microbes only persist for 1-2 weeks; I'd be surprised if most mixed diet eaters actually consume say yogurt with that frequency.

I'm particularly struck that it's so high for vegetarian vs. vegan. Indeed, it sounds plausible that dairy-associated bacteria drive. Still, it implies that a very high fraction of the vegetarians have recently eaten dairy-based foods; unsure if that's actually true.

Answer: We thank the reviewer for their in-depth comments. Indeed, dairy-associated microbes such as *Streptococcus thermophilus* were among the most predictive in distinguishing between vegans and vegetarians (**Figure 4b**). Moreover, we find the highest cumulative relative abundance of dairy SGBs in vegetarians (even higher than in omnivores, as can be seen in **Figure 4g**). We don't find this to be out of the realm of possibility since vegetarianism is distinguished from veganism almost purely on the basis of dairy consumption, since both groups do not eat meat but do eat fruits and vegetables. Very anecdotally, we find a common reason given by vegetarians for not being vegan is their love for and inability to give up cheese.

To support this, we further looked, in this revised manuscript, at responses from the FFQs for frequency of consumption of dairy products (milk, yogurt, cheese, butter, other dairy) in both UK and US PREDICT3 cohorts. We find that 96% of omnivores and 90% of vegetarians consume dairy at least once per week (see figure below, which we've added to the manuscript as **Supplementary Figure 3**). Both omnivores and vegetarians report very similar (though not identical) dairy consumption frequencies, with omnivores reporting more consumption at the higher frequencies (FFQ responses “2 or more times a day”, “once a day”, and “5-6 times a week”). Even if we consider only fermented dairy products (yogurt, cheeses; see figure below),

we still find that the vast majority of omnivores (90%) and vegetarians (84%) consume fermented dairy products at least once a week. The biggest difference in consumption of fermented dairy is that there is less consumption on a daily basis in favor of more consumption during the week. Interestingly, we observed that vegetarians consume fermented dairy more frequently on a daily basis than omnivores, which we think fits nicely with what we know anecdotally about vegetarian dairy consumption. Regardless of whether we consider all dairy products or only fermented dairy, we conclude that both omnivores and vegetarians consume dairy often enough to support our findings that dairy-associated bacteria drive our classification ability. We added these findings to the manuscript under the **Results** section “**Microbial species shared between the gut and food differ across diet patterns**” as follows:

“To lend more support to this hypothesis, we assessed omnivore and vegetarian frequency of dairy consumption (milk, yogurt, cheese, butter, other dairy) according to the FFQS in both P3 UK22A and P3 US22A cohorts. We found that 96% of omnivores and 90% of vegetarians consume dairy at least once per week (**Supplementary Figure 3**) and both diet patterns report very similar (though not identical) dairy consumption frequencies, with omnivores consuming dairy at the higher frequencies (FFQ responses “2 or more times a day”, “once a day”, and “5-6 times a week”). Even if we consider only fermented dairy products (yogurt and cheeses; **Supplementary Figure 3**), we still find that the vast majority of omnivores (90%) and vegetarians (84%) consume fermented dairy products at least once per week. Thus, we conclude that, while some microbes signature of diets that include dairy consumption could be selected to help digest dairy, others could be present in the gut microbiome as transient members derived from dairy foods themselves.”

Furthermore, it should be noted that, in both the UK and US, dairy consumption is high (OECD and Food and Agriculture Organization of the United Nations 2021), consistent with public health recommendations in both countries advocating daily consumption of dairy as part of a healthy balanced diet (Public Health England. 2016; U.S. Department of Agriculture and U.S. Department of Health and Human Services. 2020). Due to the absence of meat-based food groups in vegetarian diets, it is likely that dairy intake in this group is high to account for the lack of meat-based sources of protein. Based on the results we have presented in combination with population intakes outlined, we are confident that our sample likely consumes dairy on a regular basis, and that this is reflected in our findings.

Supplementary Figure 3 Frequency of dairy consumption across omnivores and vegetarians in P3 UK22A and P3 US22A according to FFQs. a Percentage (y-axis) of omnivores and vegetarians (x-axis) that consume dairy products (milk, yogurt, cheese, butter, or other dairy) between “two or more times per day” to “irregularly”. The consumption frequency categories were given by the FFQs. Percentages within the bar plots indicate the dairy consumption prevalence of that diet pattern in that consumption category. **b** Same as in a, but considering only fermented dairy products (yogurt and cheeses).

Minor critiques:

* L18 and L483 -- I'd be more cautious about the associations between diet-associated microbes and health. For example, Bilophila may not be playing causal roles in CRC or cardio metabolic disease; rather, it may serve as a biomarker of red meat intake, which in turn is responsible for deleterious effects on health. I also did not notice epidemiological analyses that attempt to quantify the distinct effects of specific microbes and diet vis a vis health (e.g. treating diet as a confounding variable when associating Bilophila to a given disease). Thus, I'd be more circumspect about implications between say probiotic/therapeutic potential of microbes and health.

Answer: We thank the reviewer for pointing out our misphrasing. We have added additional context in the discussion to make this point clearer, but hope that the reviewer understands our

word limit in the abstract and that is this part, we mean to purely list our results without taking a side as to what is a direct or indirect cause of disease. The discussion now reads as follows:

“Aside from SCFAs, IBD has also been linked with increases in bile acid concentrations (Vich Vila et al. 2023), whilst CRC has been linked with red meat consumption (Lescinsky et al. 2022; Chao et al. 2005), in particular with increases in *A. putredinis* and *B. wadsworthia* and decreases in gut commensals such as *S. thermophilus* (Feng et al. 2015). Nonetheless, whether these microbes are directly associated with disease or indirectly due to their links with meat consumption would require further longitudinal study to elucidate cause and effect.”

* L126 -- If the data are available, it could be interesting to report how much of differences in richness could be associated with differences in transit time or Bristol stool score?

Answer: We thank the reviewer for their suggestion and note here that the associations between richness and transit time or Bristol stool score was the subject of one of our earlier publications (Asnicar, Leeming, et al. 2021) on PREDICT1, the only cohort for which this metadata is available. In that study, we found richness to be significantly different only between Bristol stool types 3 and 4 and between types 3 and 6. Gut transit time showed a positive correlation with richness. However, we found the best predictor of transit time to be overall microbial composition as opposed to richness. Since we explored this topic extensively in our earlier work and since the current manuscript is already quite extensive on the topic of diet patterns, we would prefer to keep these topics separate and hope the reviewer agrees with this sentiment.

* L178 -- Ref to Fig 1b,h likely meant Fig 3?

Answer: Indeed, we thank the reviewer for catching this mistake which we have now corrected as suggested.

* L277 -- This may be a critique for a paper in preparation (Asnicar et al., In prep), but how much of a microbe's health rank is actually being mediated by its prediction of dietary pattern (the latter variable actually being the causal variable for health)? To get at this, it would be interesting to do a correlation between 'Health Rank' and how strongly a given microbe predicts say a Vegan vs. Mixed Diet.

Answer: This is indeed the subject of Asnicar et al. (now under review at *Nature*). There, we've calculated, on the one hand, microbial rankings based on cardiometabolic health (which we used here) and, on the other hand, a separate scoring based on various diet indices (hPDI, HEI, aMED, etc). There we show that there is some, though no perfect, concordance between these ranks (Spearman's correlation = 0.72). We think this is reflected quite nicely in our Figures 3e,k and Figure 4e, which show that, while many of the most predictive SGBs have a ranking, not all of them do. So also in our paper it is apparent that some, but not all, aspects of health might be mediated by diet.

* L313 -- I'd be curious (and fine to place in supplement) examples of associations that cannot be predicted from FFQ. With datasets of this size, are there associations between all SGPs and foods?

Answer: The reviewer brings up a good point. In fact, in our machine learning based on the entire FFQs, we see exactly that: while some SGBs are associated with the FFQs, not all of them are. This can be seen in **Figures 3f,i** and **Figure 4f (Supplementary Tables 15-17)**. Here, while some SGBs are more predictable with FFQs, like *Lawsonibacter assaccharolyticus* with an AUC of 0.78, which our previous research has shown has tight links with coffee consumption (Asnicar, Berry, et al. 2021), other SGBs have AUCs as low as 0.55 (**Supplementary Tables 14-16**). So from this, we would conclude that datasets like this don't necessarily mean that all associations are going to be predictive, which hopefully means that what we are capturing is indeed biologically relevant and not simply a methodological artifact.

* Fig 4J - Color bar is hard to read. White, it seems, is a middle score? Or, does it mean missing? I suggest a color gradient that doesn't move between colors and have white in the middle in order to read more easily

Answer: We thank the reviewer for their helpful input on the color gradient - we have now chosen a new gradient within just one color (blue; **Figure 4j**) and hope the reviewer finds it to better convey our data. White indicates "missing" in the sense that that was not a prevalent SGB in that particular food category. We have added this explanation to the figure legend, which now reads:

"Heatmap (log10 scale) of the prevalence of the 20 most common food SGBs across food samples, grouped into the three major food categories meat, dairy, fruits and vegetables to indicate which food group each SGB is likely a signature of. White/blank boxes indicate that SGB was not prevalent in that particular food category."

* I find it interesting to see that *Prevotella*, which has previously been strongly linked to high fiber diets across a number of global studies, does not appear to be one of the distinguishing taxa between the mixed and more plant-rich diets. Presuming there is room in the discussion for it, I'd be curious if these findings call into question that association.

Answer: We thank the reviewer for their insightful suggestion. We have now dedicated some sentences and space on this topic to the **Discussion** as follows:

Interestingly, we did not identify species in the *Segatella copri* (previously *Prevotella copri*) complex (Blanco-Míguez, Gálvez, et al. 2023) as a strong signature of vegetarian or vegan diets (only *S. copri* clade A was overrepresented in vegetarian but not vegan compared to omnivore gut microbiomes; **Supplementary Table 10**), despite its hypothesized role in distinguishing between westernized and non-westernized populations due to modern shifts away from high fiber diets (Tett et al. 2019; Blanco-Míguez, Gálvez, et al. 2023). There could be several reasons for our observation, such as the low prevalence of *S. copri* among westernized populations in general, such as those

analyzed here, or the diverse metabolic capabilities across this species complex, whose associations with various health metrics and lifestyles that go beyond diet we are just beginning to parse (Huang et al. 2024; Blanco-Míguez, Gálvez, et al. 2023).

* Do the authors have any information on how long people have been on a given diet? Such information could explain why some of the participants don't classify into their reported groups. Answer: Sadly, we don't have this information, though participants were asked about "long-term" dietary patterns, so presumably someone who has "tried out" veganism on and off over their lifetime but mainly consumed an omnivorous diet should have reported as "omnivore", but the reviewer brings up very good reasoning that we just cannot easily prove or disprove with our data. Nonetheless, we bring up this hypothesis in the **Results** section, where we mention the machine-learning-based diet pattern classification results as follows:

"Because our questionnaire-based data did not log when diet patterns may have been switched and instead asked participants about long-term dietary patterns, we hypothesize that the non-perfect classification might be due to individuals who switched diet patterns recently and some associations may actually be stronger than what we observed."

References

- Asnicar, Francesco, Sarah E. Berry, Ana M. Valdes, Long H. Nguyen, Gianmarco Piccinno, David A. Drew, Emily Leeming, et al. 2021. "Microbiome Connections with Host Metabolism and Habitual Diet from 1,098 Deeply Phenotyped Individuals." *Nature Medicine* 27 (2): 321–32.
- Asnicar, Francesco, Emily R. Leeming, Eirini Dimidi, Mohsen Mazidi, Paul W. Franks, Haya Al Khatib, Ana M. Valdes, et al. 2021. "Blue Poo: Impact of Gut Transit Time on the Gut Microbiome Using a Novel Marker." *Gut* 70 (9): 1665–74.
- Asnicar, Francesco, Andrew Maltez Thomas, Andrea Passerini, Levi Waldron, and Nicola Segata. 2024. "Machine Learning for Microbiologists." *Nature Reviews. Microbiology* 22 (4): 191–205.
- Beghini, Francesco, Lauren J. McIver, Aitor Blanco-Míguez, Leonard Dubois, Francesco Asnicar, Sagun Maharjan, Ana Mailyan, et al. 2021. "Integrating Taxonomic, Functional, and Strain-Level Profiling of Diverse Microbial Communities with Biobakery 3." *eLife* 10:1–42.
- Birmingham, Kate M., Sophie Stensrud, Francesco Asnicar, Ana M. Valdes, Paul W. Franks, Jonathan Wolf, George Hadjigeorgiou, et al. 2023. "Exploring the Relationship between Social Jetlag with Gut Microbial Composition, Diet and Cardiometabolic Health, in the ZOE PREDICT 1 Cohort." *European Journal of Nutrition* 62 (8): 3135–47.
- Berry, Sarah E., Ana M. Valdes, David A. Drew, Francesco Asnicar, Mohsen Mazidi, Jonathan Wolf, Joan Capdevila, et al. 2020. "Human Postprandial Responses to Food and Potential for Precision Nutrition." *Nature Medicine* 26 (6): 964–73.
- Blanco-Míguez, Aitor, Francesco Beghini, Fabio Cumbo, Lauren J. McIver, Kelsey N. Thompson, Moreno Zolfo, Paolo Manghi, et al. 2023. "Extending and Improving Metagenomic Taxonomic Profiling with Uncharacterized Species Using MetaPhlAn 4."

- Nature Biotechnology*, February. <https://doi.org/10.1038/s41587-023-01688-w>.
- Blanco-Míguez, Aitor, Eric J. C. Gálvez, Edoardo Pasolli, Francesca De Filippis, Lena Amend, Kun D. Huang, Paolo Manghi, et al. 2023. "Extension of the Segatella Copri Complex to 13 Species with Distinct Large Extrachromosomal Elements and Associations with Host Conditions." *Cell Host & Microbe* 31 (11): 1804–19.e9.
- Carmel, Ralph. 2005. "Folic Acid." In *Modern Nutrition in Health and Disease*, edited by Shils M, Shike M, Ross A, Caballero B, Cousins RJ, 11th edition:470–81. Lippincott Williams & Wilkins.
- Chao, Ann, Michael J. Thun, Cari J. Connell, Marjorie L. McCullough, Eric J. Jacobs, W. Dana Flanders, Carmen Rodriguez, Rashmi Sinha, and Eugenia E. Calle. 2005. "Meat Consumption and Risk of Colorectal Cancer." *JAMA: The Journal of the American Medical Association* 293 (2): 172–82.
- Chassagnole, C., B. Rais, E. Quentin, D. Fell, D. Fell, and J. Mazat. 2001. "An Integrated Study of Threonine-Pathway Enzyme Kinetics in Escherichia Coli." *Biochemical Journal* 356 Pt 2 (June):415–23.
- Crécy-Lagard, Valérie de, Basma El Yacoubi, Rocío Díaz de la Garza, Alexandre Noiriél, and Andrew D. Hanson. 2007. "Comparative Genomics of Bacterial and Plant Folate Synthesis and Salvage: Predictions and Validations." *BMC Genomics* 8 (July):245.
- DerSimonian, R., and N. Laird. 1986. "Meta-Analysis in Clinical Trials." *Controlled Clinical Trials* 7 (3): 177–88.
- Diener, Christian, and Sean M. Gibbons. 2024. "Metagenomic Estimation of Dietary Intake from Human Stool." *bioRxiv : The Preprint Server for Biology*, February. <https://doi.org/10.1101/2024.02.02.578701>.
- Dosselaere, F., and J. Vanderleyden. 2001. "A Metabolic Node in Action: Chorismate-Utilizing Enzymes in Microorganisms." *Critical Reviews in Microbiology* 27 (2): 75–131.
- Feng, Qiang, Suisha Liang, Huijue Jia, Andreas Stadlmayr, Longqing Tang, Zhou Lan, Dongya Zhang, et al. 2015. "Gut Microbiome Development along the Colorectal Adenoma-Carcinoma Sequence." *Nature Communications* 6 (March):6528.
- Ghensi, Paolo, Paolo Manghi, Moreno Zolfo, Federica Armanini, Edoardo Pasolli, Mattia Bolzan, Alberto Bertelle, et al. 2020. "Strong Oral Plaque Microbiome Signatures for Dental Implant Diseases Identified by Strain-Resolution Metagenomics." *Npj Biofilms and Microbiomes* 6 (1): 47.
- Green, Ralph, Lindsay H. Allen, Anne-Lise Bjørke-Monsen, Alex Brito, Jean-Louis Guéant, Joshua W. Miller, Anne M. Molloy, et al. 2017. "Vitamin B12 Deficiency." *Nature Reviews. Disease Primers* 3 (June):17040.
- Handzlik, Michal K., and Christian M. Metallo. 2023. "Sources and Sinks of Serine in Nutrition, Health, and Disease." *Annual Review of Nutrition* 43 (August):123–51.
- Hoch, J. A., and E. W. Nester. 1973. "Gene-Enzyme Relationships of Aromatic Acid Biosynthesis in Bacillus Subtilis." *Journal of Bacteriology* 116 (1): 59–66.
- Huang, Kun D., Lena Amend, Eric J. C. Gálvez, Till-Robin Lesker, Romulo de Oliveira, Agata Bielecka, Aitor Blanco-Míguez, et al. 2024. "Establishment of a Non-Westernized Gut Microbiota in Men Who Have Sex with Men Is Associated with Sexual Practices." *Cell Reports. Medicine* 5 (3): 101426.
- Lee, Karla A., Andrew Maltez Thomas, Laura A. Bolte, Johannes R. Björk, Laura Kist Ruijter, Federica Armanini, Francesco Asnicar, et al. n.d. "Cross-Cohort Gut Microbiome Associations with Immune Checkpoint Inhibitor Response in Advanced Melanoma." Springer US. <https://doi.org/10.1038/s41591-022-01695-5>.
- Lescinsky, Haley, Ashkan Afshin, Charlie Ashbaugh, Catherine Bisignano, Michael Brauer, Giannina Ferrara, Simon I. Hay, et al. 2022. "Health Effects Associated with Consumption of Unprocessed Red Meat: A Burden of Proof Study." *Nature Medicine* 28 (10): 2075–82.
- Manghi, Paolo, Aitor Blanco-Míguez, Serena Manara, Amir NabiNejad, Fabio Cumbo,

- Francesco Beghini, Federica Armanini, et al. 2023. "MetaPhlAn 4 Profiling of Unknown Species-Level Genome Bins Improves the Characterization of Diet-Associated Microbiome Changes in Mice." *Cell Reports* 42 (5): 112464.
- Mazidi, Mohsen, Ana M. Valdes, Jose M. Ordovas, Wendy L. Hall, Joan C. Pujol, Jonathan Wolf, George Hadjigeorgiou, et al. 2021. "Meal-Induced Inflammation: Postprandial Insights from the Personalised REsponses to Dietary Composition Trial (PREDICT) Study in 1000 Participants." *The American Journal of Clinical Nutrition* 114 (3): 1028–38.
- Nagy, P. L., A. Marolewski, S. J. Benkovic, and H. Zalkin. 1995. "Formyltetrahydrofolate Hydrolase, a Regulatory Enzyme That Functions to Balance Pools of Tetrahydrofolate and One-Carbon Tetrahydrofolate Adducts in Escherichia Coli." *Journal of Bacteriology* 177 (5): 1292–98.
- OECD and Food and Agriculture Organization of the United Nations. 2021. "7. Dairy and Dairy Products." In *OECD-FAO Agricultural Outlook 2021-2030*. OECD-FAO Agricultural Outlook. OECD Publishing. <https://doi.org/10.1787/19428846-en>.
- Public Health England. 2016. "The Eatwell Guide. Helping You Eat a Healthy, Balanced Diet." Public Health England. <https://www.gov.uk/government/publications/the-eatwell-guide>.
- Richards, Thomas A., Joel B. Dacks, Samantha A. Campbell, Jeffrey L. Blanchard, Peter G. Foster, Rima McLeod, and Craig W. Roberts. 2006. "Evolutionary Origins of the Eukaryotic Shikimate Pathway: Gene Fusions, Horizontal Gene Transfer, and Endosymbiotic Replacements." *Eukaryotic Cell* 5 (9): 1517–31.
- Rothschild, Daphna, Omer Weissbrod, Elad Barkan, Alexander Kurilshikov, Tal Korem, David Zeevi, Paul I. Costea, et al. 2018. "Environment Dominates over Host Genetics in Shaping Human Gut Microbiota." *Nature* 555 (7695): 210–15.
- Tarallo, Sonia, Giulio Ferrero, Gaetano Gallo, Antonio Francavilla, Giuseppe Clerico, Alberto Realis Luc, Paolo Manghi, et al. 2019. "Altered Fecal Small RNA Profiles in Colorectal Cancer Reflect Gut Microbiome Composition in Stool Samples." *mSystems* 4 (5). <https://doi.org/10.1128/mSystems.00289-19>.
- Tett, Adrian, Kun D. Huang, Francesco Asnicar, Hannah Fehlner-Peach, Edoardo Pasolli, Nicolai Karcher, Federica Armanini, et al. 2019. "The Prevotella Copri Complex Comprises Four Distinct Clades Underrepresented in Westernized Populations." *Cell Host & Microbe* 26 (5): 666–79.
- Thomas, Andrew Maltez, Paolo Manghi, Francesco Asnicar, Edoardo Pasolli, Federica Armanini, Moreno Zolfo, Francesco Beghini, et al. 2019. "Metagenomic Analysis of Colorectal Cancer Datasets Identifies Cross-Cohort Microbial Diagnostic Signatures and a Link with Choline Degradation." *Nature Medicine* 25 (4): 667–78.
- U.S. Department of Agriculture and U.S. Department of Health and Human Services. 2020. "Dietary Guidelines for Americans, 2020-2025. 9th Edition." December 2020. <https://health.gov/our-work/nutrition-physical-activity/dietary-guidelines/current-dietary-guidelines>.
- Valles-Colomer, Mireia, Paolo Manghi, Fabio Cumbo, Giulia Masetti, Federica Armanini, Francesco Asnicar, Aitor Blanco-Miguez, et al. 2024. "Neuroblastoma Is Associated with Alterations in Gut Microbiome Composition Subsequent to Maternal Microbial Seeding." *EBioMedicine* 99 (104917): 104917.
- Vich Vila, Arnau, Shixian Hu, Sergio Andreu-Sánchez, Valerie Collij, Bernadien H. Jansen, Hannah E. Augustijn, Laura A. Bolte, et al. 2023. "Faecal Metabolome and Its Determinants in Inflammatory Bowel Disease." *Gut* 72 (8): 1472–85.
- Watanabe, Fumio, Yukinori Yabuta, Yuri Tanioka, and Tomohiro Bito. 2013. "Biologically Active Vitamin B12 Compounds in Foods for Preventing Deficiency among Vegetarians and Elderly Subjects." *Journal of Agricultural and Food Chemistry* 61 (28): 6769–75.
- Weber, Michael, and Thilo M. Fuchs. 2022. "Metabolism in the Niche: A Large-Scale Genome-Based Survey Reveals Inositol Utilization To Be Widespread among Soil,

Commensal, and Pathogenic Bacteria.” *Microbiology Spectrum* 10 (4): e0201322.

Willett, Walter, Johan Rockström, Brent Loken, Marco Springmann, Tim Lang, Sonja Vermeulen, Tara Garnett, et al. 2019. “Food in the Anthropocene: The EAT–Lancet Commission on Healthy Diets from Sustainable Food Systems.” *The Lancet* 393 (10170): 447–92.

Nature Microbiology manuscript NMICROBIOL-24030957A

“Gut microbiome signatures of vegan, vegetarian and omnivore diets and associated health outcomes across 21,561 individuals”

Response to Editorial and Reviewers’ comments

In this letter we report the text from the editor and the reviewers with black text and our response in green. We are also submitting the paper with “track changes”.

Comments from the Editor:

Dear Dr. Segata,

Thank you for submitting your revised manuscript "Microbiome signatures of vegan, vegetarian and omnivore diets and links to health outcomes: a multi-population study of 21,561 individuals" (NMICROBIOL-24030957A). It has now been seen by the original referees and their comments are below. The reviewers find that the paper has improved in revision, and therefore we'll be happy in principle to publish it in Nature Microbiology, pending minor revisions to satisfy the referees' final requests and to comply with our editorial and formatting guidelines.

We are now performing detailed checks on your paper and will send you a checklist detailing our editorial and formatting requirements in a few weeks. Please do not upload the final materials and make any revisions until you receive this additional information from us.

Thank you again for your interest in Nature Microbiology Please do not hesitate to contact me if you have any questions.

Sincerely,
Paula

Paula Jauregui, PhD
Senior Editor
Nature Microbiology

Answer: We thank the editor and reviewers for taking the time to consider our manuscript and for their thoughtful comments. We are pleased to have addressed all the reviewers' concerns in the first revision round, though it seems that there was some misunderstanding pertaining to the MEDI analysis that we added to the manuscript. Reviewer #1 has clarified in this revision round that they expected us to only discuss this tool and not necessarily use it. Since Reviewer #1 is an author of this tool and appears to be satisfied with us having adopted it, we have decided to keep this analysis in the manuscript, also because we invested a significant amount of time implementing it in the first revision round. Nonetheless, we have taken their comments concerning potential limitations of our implementation of the tool to heart and mentioned these caveats in this version of the manuscript. Reviewer #2 also expressed concern with how we adopted this tool, saying they would have expected us to run our machine learning analysis on the MEDI estimates. However, such ML-based analysis would be more an evaluation of the overall performance of the tool (rather than a way to prove or disprove any hypotheses set out in our work), which we believe should be tackled by the developers in their upcoming peer-reviewed publication or by other benchmarking studies. Therefore, we opted to leave the MEDI analysis as is and leave any further validation of the tool to its authors and future adopters, and instead toned-down the wording that reviewer #2 found troubling to reflect the potential caveats mentioned by both reviewers. More detailed replies to reviewer comments are given below. We hope that these textual changes adequately address the editor's and reviewers' concerns and hope that this work is now suitable for publication in *Nature Microbiology*.

Comments from Reviewer #1 (Remarks to the Author):

I thank the authors for their thorough response to all reviewer concerns. The code availability section is now acceptable. The functional analysis adds a lot, and I appreciate their efforts. I commend them on an excellent manuscript. I have no further concerns, just a minor comment/clarification below.

Answer: We thank the reviewer for their kind comments and are very happy to have fully addressed their concerns.

I was surprised the authors ran the MEDI analysis, which is really great and truly above and beyond what I was expecting (especially given that its still only a preprint, as they point out)! I'm sorry if I was unclear in my original review. I saw your current work as another attempt to build a data-driven method for classifying diet from microbiome compositional data for samples that lack FFQs or dietary questionnaire data. I have no doubt in the accuracy of your FFQs, and I think questionnaires are still the gold standard for dietary assessment. Data-driven methods are noisy, but they can provide additional information when we lack assessment data (and possibly, they might help correct certain biases inherent to questionnaires). When I made my comment, I was only asking the authors to discuss these alternative data-driven approaches and how these methods might be complementary, moving forward.

At this point, I might as well out myself as an author of the MEDI paper (Sean Gibbons). It's great to see that animal DNA is more prevalent in the omnivores. As for the other results, they make total sense given our experience. MEDI profiles are pretty sparse (I think of them like single-cell sequencing data). In your results, you may want to discuss power, in the context of not detecting animal DNA (or any other food item) in samples where you expect it. In our preprint, we find that you need at least 10 reads per million to get reliable detection of a given food item, and <0.1% of stool reads are diet-derived. Often, we find dietary profiles for metagenomes with fewer than ~30 million reads are too noisy to work with on a sample-by-sample basis (i.e., too much sparsity to interpret data coming from a single sample). It looks like your metagenomes were sequenced to depths of 10-20 million reads, which is below this depth level (we also see higher sparsity at these depths). In lower depth contexts, we see better results when pooling samples within diet groupings and looking at differences across groups (similar to what you did with animal DNA across the diets, which seems to work ok). We've also made some tweaks to the method during the review process that improves animal DNA detection (the final version of the method will have some improvements from what's presented in the preprint), but I don't expect you to rerun anything at this point (what you've already done is beyond what I expected). Anyway, thanks for trying out MEDI. Your results are consistent with what we're seeing. And I'm excited to explore if combining your microbiome signatures with our MEDI estimates might further improve data-driven dietary assessment/prediction (when lacking questionnaire data, which can be expensive to obtain and time consuming for study participants).

Answer: We thank the reviewer for clarifying their earlier comment, since it seems we misunderstood it. We are happy to hear that the reviewer is satisfied with us having adopted MEDI in our work and are encouraged to keep these results in the manuscript. We understand that this tool is still in its developmental stages and that future modifications could change our results. We highlight some shortcomings in our implementation of the tool that both Reviewer #1 and #2 mentioned in the Results and Methods sections as follows (please note that we had to cut down the word count by 40% so this is reflected throughout):

Results: In addition to participants' overall dietary habits, the ZOE PREDICT cohorts also included data on habitual consumption of over 150 single foods per individual, obtained from validated quantitative food frequency questionnaires (FFQs; **Methods**). Dietary patterns were partially confirmed by DNA-based detection of food in the stool microbiome¹⁴ that, however, would require greater sequencing depth to be used for this goal (**Methods**).

Methods: To further support the FFQs, we tested the Metagenomic Estimation of Dietary Intake (MEDI) tool¹⁴, which uses food DNA in gut metagenomes to estimate and quantify food consumption (<https://github.com/Gibbons-Lab/medi>). We assessed how a MEDI-based classification of diet patterns would perform versus an FFQ-based classification, using what participants self-reported as the ground truth (only participants from P1, P3 22UKA, and P3 22USA were considered, since these are the only cohorts with FFQs and all three diet patterns). To do so, we classified any sample in which MEDI found animal DNA

as a non-vegan sample and any sample in which no animal DNA was found as a vegan sample. Similarly, we classified any sample whose FFQ reported the consumption of any animal product as a non-vegan sample and vice versa. Indeed, we found a lower prevalence of animal DNA among vegans vs non-vegans using the MEDI classification (**Extended Data Figure 7; Supplementary Table 23**), which highlights this tool's potential application in studies lacking data on overall dietary patterns. Compared to FFQs, however, we found MEDI unable to perform similarly well in predicting participants' diet patterns (chi-squared test, $p < 0.001$). While accurate thresholding of MEDI-derived statistics could improve performance, a deeper sequencing depth may be needed to substantially increase the tool's reliability by capturing a greater amount of food DNA, which is generally sparse in fecal samples. Since FFQs remain the gold standard and given the focus of our work on long versus short-term dietary patterns, we opted to base any analyses using food consumption data on the FFQs that have been extensively validated over time and in publications^{10,11} and refer researchers to adopt a MEDI-based approach for studies lacking proper FFQs or as an additional validation tool.

We hope the reviewer finds this description to both accurately reflect our results and highlight their shortcomings.

Comments from Reviewer #2 (Remarks to the Author):

Overall, I am satisfied with the authors' responses to my first round of critiques. Kudos to them for a thorough response.

Answer: We thank the reviewer for their kind comments and are happy to have addressed all their concerns with our previous response.

I do have some comments though in regards to the authors' responses to Reviewer #1. I'll note that I typically do not engage in responses to other reviewers to minimize workload for the authors; however, in this case, I have some concerns with the authors' use/interpretation of MEDI, which could have an inadvertent and disproportionate effect on adoption of the tool since it is so young (in fact, I believe it is still a pre-print).

In short, I have concerns about the conclusion: "this tool nonetheless performed significantly worse than the FFQs in predicting participants' diet patterns (Chi-squared test, $p < 0.001$), with MEDI misclassifying 77% of participants."

My understanding is that Reviewer #1 wasn't asking for MEDI to be compared against the FFQ in terms of predicting diet. I did not see them question the veracity of FFQ. Their question was instead how MEDI compared against microbiome-based markers in terms of predicting diet. ("The thrust of this paper is to look for bacterial taxonomic biomarkers of dietary intake, but

there are a few existing approaches to dietary assessment directly from data that are not discussed by the authors. It would be useful to compare and contrast their results to these other methods. For example, prior work has proposed using ...direct detection of whole-genome food-related DNA in metagenomes (<https://www.biorxiv.org/content/10.1101/2024.02.02.578701v1>)").

Furthermore, it seems unfair to consider simply presence/absence of total animal DNA as a decision rule for diet when the microbiome-based markers are using far more powerful ML techniques (e.g. RFs) to predict diet. If the authors choose to include the MEDI analyses in their paper, it seems fairer to ask how the same ML-based techniques used to predict diet perform when applied to the MEDI data.

Furthermore, I was struck by what the authors also noticed, which was that there was a much lower abundance of animal DNA in the vegetarians and vegans. I find it quite promising that MEDI could at the very least help confirm whether someone is indeed vegetarian or vegan. (I do see though that it doesn't look as promising for identifying omnivores from a single sample.)

In summary, my concern is that the author's new MEDI results could be mis-interpreted as a strongly negative result, when in reality, what was carried out was not designed to be a full and thorough validation of MEDI's potential and accuracy. I think it would be regrettable to inadvertently quash interest in this tool before it has had a chance to be fully explored and tested by the community.

Answer: We thank the reviewer for taking the time to assess our work and also take the other reviewer's comments into consideration. Reviewer #1 has also clarified their position, which we find very helpful, since we seem to have misunderstood their previous comments. Our aim with including the ML analysis that we already performed was to show that we see substantial differences in the diet patterns and to further explore what these differences are and interpret them biologically. For this reason, our Discussion is tailored to explaining the gut microbial differences between the 3 diet patterns explored here and how these could have arisen (diet-driven selection, potential food-to-gut transfer, etc.). Discussing tools that could be used in other studies to identify biomarkers for diet patterns would shift the focus away from the main biological message we want to convey with this paper. We also find that conducting a ML analysis using the MEDI estimates would serve more as a validation of the tool than a validation of our results or hypotheses. While such an analysis could be the subject of future work whose focus relies mainly on this, we find that our current work does not fit this description. Nonetheless, we understand the reviewer's concerns and certainly do not intend to prematurely diminish MEDI's validity. Since Reviewer #1 has stated that they are an author of MEDI and appears satisfied that we did use their tool, despite it not being their intention, we have opted to keep this analysis in the paper, also because we invested significant time in it. However, we acknowledge the caveats of our adoption of this method and have described these in the Results and Methods sections, so that other researchers may learn and improve upon it, and have also toned down some of the wording that the reviewer found troubling. The text now

reads as follows (please note that we had to cut down the word count by 40% so this is reflected throughout):

Results: In addition to participants' overall dietary habits, the ZOE PREDICT cohorts also included data on habitual consumption of over 150 single foods per individual, obtained from validated quantitative food frequency questionnaires (FFQs; **Methods**). Dietary patterns were partially confirmed by DNA-based detection of food in the stool microbiome¹⁴ that, however, would require greater sequencing depth to be used for this goal (**Methods**).

Methods: To further support the FFQs, we tested the Metagenomic Estimation of Dietary Intake (MEDI) tool¹⁴, which uses food DNA in gut metagenomes to estimate and quantify food consumption (<https://github.com/Gibbons-Lab/medi>). We assessed how a MEDI-based classification of diet patterns would perform versus an FFQ-based classification, using what participants self-reported as the ground truth (only participants from P1, P3 22UKA, and P3 22USA were considered, since these are the only cohorts with FFQs and all three diet patterns). To do so, we classified any sample in which MEDI found animal DNA as a non-vegan sample and any sample in which no animal DNA was found as a vegan sample. Similarly, we classified any sample whose FFQ reported the consumption of any animal product as a non-vegan sample and vice versa. Indeed, we found a lower prevalence of animal DNA among vegans vs non-vegans using the MEDI classification (**Extended Data Figure 7; Supplementary Table 23**), which highlights this tool's potential application in studies lacking data on overall dietary patterns. Compared to FFQs, however, we found MEDI unable to perform similarly well in predicting participants' diet patterns (chi-squared test, $p < 0.001$). While accurate thresholding of MEDI-derived statistics could improve performance, a deeper sequencing depth may be needed to substantially increase the tool's reliability by capturing a greater amount of food DNA, which is generally sparse in fecal samples. Since FFQs remain the gold standard and given the focus of our work on long versus short-term dietary patterns, we opted to base any analyses using food consumption data on the FFQs that have been extensively validated over time and in publications^{10,11} and refer researchers to adopt a MEDI-based approach for studies lacking proper FFQs or as an additional validation tool.